# Molecular Basis for the Involvement of Mammalian Serum Albumin in the AGE/RAGE Axis: A Comprehensive Computational Study

**DOI:** 10.3390/ijms25063204

**Published:** 2024-03-11

**Authors:** Daria A. Belinskaia, Richard O. Jenkins, Nikolay V. Goncharov

**Affiliations:** 1Sechenov Institute of Evolutionary Physiology and Biochemistry, Russian Academy of Sciences, pr. Torez 44, 194223 St. Petersburg, Russia; 2Leicester School of Allied Health Sciences, De Montfort University, The Gateway, Leicester LE1 9BH, UK; 3Research Institute of Hygiene, Occupational Pathology and Human Ecology, Bld.93 p.o. Kuz’molovsky, 188663 St. Petersburg, Russia

**Keywords:** carboxymethyl-lysine, diabetes mellitus, fatty acids, glycated albumin, macromolecular docking, molecular dynamics, receptor for advanced glycation end products

## Abstract

In mammals, glycated serum albumin (gSA) contributes to the pathogenesis of many metabolic diseases by activating the receptors (RAGE) for advanced glycation end products (AGEs). Many aspects of the gSA–RAGE interaction remain unknown. The purpose of the present paper was to study the interaction of glycated human albumin (gHSA) with RAGE using molecular modeling methods. Ten models of gHSA modified with different lysine residues to carboxymethyl-lysines were prepared. Complexes of gHSA–RAGE were obtained by the macromolecular docking method with subsequent molecular dynamics simulation (MD). According to the MD, the RAGE complexes with gHSA glycated at Lys233, Lys64, Lys525, Lys262 and Lys378 are the strongest. Three-dimensional models of the RAGE dimers with gHSA were proposed. Additional computational experiments showed that the binding of fatty acids (FAs) to HSA does not affect the ability of Lys525 (the most reactive lysine) to be glycated. In contrast, modification of Lys525 reduces the affinity of albumin for FA. The interspecies differences in the molecular structure of albumin that may affect the mechanism of the gSA–RAGE interaction were discussed. The obtained results will help us to learn more about the molecular basis for the involvement of serum albumin in the AGE/RAGE axis and improve the methodology for studying cellular signaling pathways involving RAGE.

## 1. Introduction

Advanced glycation end products (AGEs) are the result of the non-enzymatic condensation of sugars with nucleic acids, proteins and lipids. AGEs can enter the body from external sources, mainly from food, or be a result of unbalanced detoxification mechanisms [1,2]. Glycation alters the structure and function of biomolecules, leading to cell dysfunction and cytotoxic effects, and that is why AGEs are called glycotoxins. Circulating AGEs bind to specific receptors (RAGE) and activate a range of signaling cascades in cells, leading to inflammation and oxidative stress. In particular, activation of the AGE–RAGE axis enhances the generation of reactive oxygen species by nicotinamide adenine dinucleotide phosphate oxidase and induces the expression of nuclear factor NF-κB, which leads to the recruitment and activation of inflammatory cells bearing S100/calgranulin and high-mobility group protein B1 (HMGB1) and triggers the inflammatory process [3]. Traditionally, the formation and toxic effects of AGEs in the human body are associated with long-term hyperglycemia caused by diabetes mellitus (DM). However, in addition to DM, glycated molecules and RAGE activation are associated with the pathophysiology of asthma, food allergies, acute renal failure, chronic obstructive pulmonary disease, polycystic ovary syndrome, Alzheimer’s disease, etc. [4,5,6].

Serum albumin is a major protein in the blood of mammals and some other classes, where its concentration is 500–700 µM. The difference between albumin and other blood proteins is that it is normally not glycated, but due to its high concentration, even a small percentage of gSA makes a significant contribution to the overall level of AGEs. It is known that glycated human albumin (gHSA) can bind to the RAGE of endothelial cells [7,8] and adipocytes [9]. In order to investigate the gSA–RAGE interaction at the molecular level, it is necessary to resolve the three-dimensional structure of the gSA–RAGE complex. This structure has not yet been resolved experimentally. However, sufficient experimental data have been collected on the AGEs’ interaction sites on the RAGE surface [10]. The accumulated information allows the application of molecular modeling methods to construct the gSA–RAGE complexes and determine the molecular basis for the involvement of serum albumin in the AGE/RAGE axis. This, in turn, will inform us about the role of gSA in the pathogenesis of DM and other metabolic disorders.

The purpose of the present research was to perform for the first time a comprehensive study of the interaction of gHSA with RAGE using molecular modeling methods. For this purpose, we set the following tasks:(1)to perform multiple macromolecular docking between gHSA and RAGE;(2)to study the conformational characteristics of the complexes using molecular dynamics (MD) simulation;(3)to determine the most likely binding sites on the surface of gHSA;(4)to construct presumptive structures of gHSA–RAGE dimers;(5)to perform a detailed comparative analysis of the primary sequences of albumin from different species in order to determine whether there are interspecific differences in the molecular characteristics of albumin that determine its interaction with RAGE.

## 2. Results

### 2.1. Choosing a Model of RAGE

RAGE is a multiligand plasma membrane receptor belonging to the immunoglobulin superfamily. In addition to AGEs, its ligand-binding domain recognizes a huge number of molecules, among which the most important are β-amyloid and macrophage antigen complex-1, HMGB1, and the S100 proteins [11]. A schematic representation of a monomer of RAGE is shown in Figure 1.

The three-dimensional structure of the RAGE monomer includes three regions: (1) the extracellular ligand-binding region containing immunoglobulin-like domain V (residues 23–119), domains C1 (residues 120–233) and C2 (residues 234–325), (2) the single transmembrane helix (residues 343–363) and (3) the short C-terminal cytoplasmic domain (residues 364–404) (Figure 1). It is known that, despite their structural diversity, AGEs (including AGE proteins) bind only to the V-domain of RAGE [8]. It is believed that the signal transduction caused by ligand binding to RAGE requires receptor oligomerization [12]. The three-dimensional structure of the albumin complex with either the monomer or the oligomer of RAGE has not yet been obtained experimentally. So, at the first stage of this study, we analyzed the data published in the literature on the interaction of RAGE with other macromolecular ligands in order to identify all the possible options for the architecture of the RAGE–gHSA complex.

The Protein Data Bank (PDB) contains several RAGE structures obtained via the methods of X-ray analysis or nuclear magnetic resonance (NMR). For further analysis, we selected those models that provide insight into the structure of the RAGE oligomer and the interaction of the V-domain of RAGE with macromolecules. Thus, the structures presented in Table 1 came into our field of view.

The complex of RAGE with S100, according to the NMR data, can be schematically described by the structure presented in Figure 2A. In this dimer, the V-domain monomers do not contact each other and the dimer of S100 is sandwiched between them. The structures 2M1K and 2MJW contain neither C1 nor C2 domains. However, when the V-domains of structure 2M1K [13] and the V-domain of structure 4LP5 (full-length human RAGE extracellular domain [12]) are aligned, it is possible to make sure that the C1 and C2 domains of the dimer of the RAGE–S100 complex do not contact each other neither. A complete model of the entire complex was proposed by Xue et al.: S100 is sandwiched between the two RAGE monomers, which contact other RAGE monomers not involved in S100 binding [16]. Thus, the whole oligomer, according to [16], consists of four RAGE monomers and two S100 monomers.

The complex of RAGE with DNA, according to X-ray analysis [15], can be schematically described by the structure shown in Figure 2B. In this complex, RAGE forms the dimer in which the V-domains do contact, but the C1-domains (and, as a consequence, the C2-domains) do not contact each other. The DNA molecule is associated with both V-domains of the dimer and partially with the C1-domains. A similar dimer structure is formed when RAGE interacts with low-molecular-weight ligands (see PDB entries 6XQ1, 6XQ3, 6XQ5, 6XQ6, 6XQ7, 6XQ8, and 6XQ9 [17]).

Xie et al. proposed the architecture for the RAGE complex with albumin [10]. A simplified scheme of this architecture is presented in Figure 2C. In this complex, the albumin molecule interacts with at least two monomers of the V-domain, which are part of the RAGE oligomer associated through the C1-domains. Although the architecture of the complex presented in Figure 2C has not yet been confirmed experimentally, we do not exclude this variant from consideration.

It interesting that, when structures 2M1K, 2MJW and 4OI7 (4OI8) are superimposed, their V-domains do not coincide spatially. Thus, the spatial organization of the RAGE dimer depends on which macromolecule it interacts with. Even the type of the S100 protein matters. Therefore, the conformation of the RAGE dimer in complex with albumin may differ greatly from the dimer structures available in the PDB database. Moreover, it is unknown how the process of albumin binding to RAGE occurs, whether albumin binds directly to the dimer or first to the monomer, and then dimerization occurs. Thus, the first reasonable stage is to simulate the interaction of gHSA with the monomer of the V-domain, then to select the most probable (strongest) complexes of these two proteins, and then to try to assemble the selected structures into a dimer and determine the most probable architecture of the gHSA–RAGE complex.

### 2.2. Building Glycated Albumin Models

The glycation process mainly targets N-terminal amino acids, the thiol groups of cysteine and the side chains of lysine and arginine. In HSA, glycation at the lysine residues has been studied the most. Moreover, the glycated lysines of lysine-rich proteins (including serum albumin) are markers of age-related pathologies [18]. Carboxymethyl-lysine (Figure 3) is one of the main reagents found in vivo in adducts with albumin [19].

The tertiary structure of albumin includes three homologous domains: I, II and III. Each of these domains contains two subdomains, A and B, consisting of six and four helices, respectively. Previously, it was established that at physiologically relevant concentrations, HSA forms weak, reversible non-covalent dimers [20]. Presumably, in the bloodstream and in extravascular fluids, there is some equilibrium between the concentration of the albumin dimers and monomers. Since extravascular fluids contain much lower levels of HSA, the equilibrium is shifted toward more monomers and fewer dimers [20]. Since albumin interacts with RAGE, both in blood vessels and in extravascular fluids, it seems reasonable to devote the present study to the binding of albumin monomers to RAGE and then, in future works, to model the binding of the dimers on the basis of the described information.

HSA molecules contains 59 lysines, and it is reasonable to suggest that only lysines located on the surface of albumin are available for interaction with RAGE. There are the following lysines on the surface of the protein: 4, 20, 41, 51, 64, 73, 93, 136, 159, 162, 174, 181, 205, 212, 225, 233, 240, 262, 274, 286, 313, 317, 323, 351, 359, 372, 378, 389, 402, 439, 444, 475, 500, 519, 524, 525, 534, 545, 557, 560, 564, 573, and 574. The review by Qiu et al. summarized the literature data, according to which HSA lysines were detected to be glycated in the blood of healthy volunteers and DM patients [21]. Of these lysines, the following residues were most often glycated: 64, 73, 137, 233, 262, 317, 378, 525, 573 and 574. Therefore, we selected these 10 lysines to construct the models of glycated HSA. In the present paper, we have studied only mono-glycated models of HSA. The advantage of molecular modeling methods is that they can be used to separately study the contribution of each glycation to the features of gSA. Determination of the single glycation contribution is important for comparing albumins of different types, for developing metabolic disease therapies (since glycation and/or binding to RAGE can affect the affinity of the protein for various pharmaceuticals), and for studying the phenotypic features of patients with mutated albumin (for example, the Canterbury 313Lys→Asn, Vanves 574Lys→Asn or Verona 570Glu→Lys mutations [22]).

Using the molecular docking procedure with subsequent MD simulation, we built ten gHSA models with one glycated lysine residue from the selected list. The procedure for preparing the gHSA models is described in more detail in Section 4.2. Figure 4 shows the resulting structures.

As expected, during the MD simulation, all the studied glycated lysines, due to their hydrophilic carboxyl group, converged to the conformations in which their side chains are maximally exposed to the solvent. In Lys317-glycated HSA, the carboxyl group of CML forms an intramolecular salt bridge with Lys313. In Lys525-glycated HSA, the glycated lysine forms a salt bridge with Arg521 and a hydrogen bond with Gln522. In the other eight gHSA models, the carboxyl group of CML does not form close contacts with the surrounding residues of HSA.

Additionally, before the macromolecular docking procedure, we studied how the glycation of the considered lysines affects the structural characteristics of gHSA. We calculated the time dependence of such protein parameters as the radius of gyration (Rg) and solvent accessible surface area (SASA), as well as the root mean square fluctuation (RMSF) of Cα atoms. The plots are presented in Appendix A (Appendix A, respectively). The Rg value characterizes the degree of protein compactness; an increase in this parameter during the simulation means an increase in the degree of protein unfolding, and vice versa. According to the obtained results, glycation of Lys64 (Appendix A), Lys73 (Appendix A), Lys137 (Appendix A), Lys262 (Appendix A) and Lys378 (Appendix A) does not lead to an increase in the Rg parameter; the protein retains its secondary and tertiary structures. In the case of Lys233 (Appendix A), Lys317 (Appendix A), Lys573 (Appendix A) and Lys574 (Appendix A), the Rg value fluctuates and then stabilizes. And only in the case of Lys525 does the Rg value increase smoothly throughout the entire simulation (Appendix A). Apparently, glycation of Lys525 leads to a more significant rearrangement of the protein globule. In the case of lysines belonging to domain IIIB (Lys525, Lys573 and Lys574), gHSA is also characterized by a higher SASA value (Appendix A) compared to lysines from the other HSA domains (Appendix A). That is, apparently, the structure of domain III is the most susceptible to conformational changes after the glycation of its amino acid residues.

Finally, the RMSF value shows which amino acid residues are the most mobile during the process of conformational changes caused by the glycation of the lysines considered. It is interesting to note that the smallest fluctuations are caused by glycation of Lys64 (Appendix A) and Lys73 (Appendix A), belonging to domain IA, as well as Lys378 (Appendix A). The latter, although it belongs to domain IIB, is located at the very border with domain IIIA. Glycation of Lys137 (Appendix A), Lys233 (Appendix A) and Lys525 (Appendix A) leads to significant fluctuations in domain IIB (in the region of amino acids 300–350, and in the case of Lys525, also in the region of amino acids 350–400). Glycation of almost all the lysines (except for Lys64 and Lys73) results in the increased mobility of the amino acids of domain III (Appendix A). Lys64 and Lys73 belong to domain IA, which topographically is the most remote from domain III.

### 2.3. Macromolecular Docking of gHSA with the V-Domain of RAGE

At the next stage, we performed macromolecular (protein–protein) docking of the prepared gHSA models with RAGE. The result of running the ZDOCK program is a set of the 10 most favorable (according to the ZDOCK scoring function, SF) structures of the complex of gHSA with the V-domain of the receptor (gHSA–V). For each gHSA model, from the set of output conformations, we picked up the most probable structures based on the following criteria. The first one is that when superimposing the docked gHSA–V complex onto PDB structure 4OI7 (crystal structure of the VC1-domain of RAGE [15]), the albumin molecule should not overlap the receptor C1-domain (that is, gHSA should not be docked at the region between the V- and C1-domains of RAGE). The second criterion is that, in the output gHSA–V complex, the glycated lysine must contact one of the experimentally obtained interaction surfaces (ISs) of RAGE described by Xie et al. [10]. The ISs (IS1, IS2 and IS3) are the regions on the surface of RAGE that bind AGE peptides and full-length glycated bovine serum albumin (gBSA). IS1 includes Leu36, Lys37, Cys38, Gly40, Ala41, Lys43, Leu49, Asn81 and Cys99; IS2 contains Arg48, Glu50, Arg98, Met102, Asn105, Gly106 and Lys107; and IS3 consists of Arg28, Arg29, Ile30, Ile91, Glu94 and Gly95.

The obtained structures optimized by energy minimization using GROMACS 2019.4 software (University of Groningen, the Netherlands) [23] are available in PDB format in the Appendix A. The result of the macromolecular docking is presented in Figure 5 and Table 2. In the case of the modification of Lys64, the CML in the gHSA–V complex interacts sterically with Lys52 and forms a salt bridge with the Arg98 of the IS2 of the receptor (Figure 5A). In the complex of RAGE with Lys73-glycated HSA, the CML binds in the vicinity of Lys107 (belongs to IS2), Glu108 and Thr109 (Figure 5B). In the case of the modification of Lys137, the CML forms a close contact with the Arg48 of IS2 (Figure 5C). In the complex of RAGE with Lys233-glycated HSA, the modified lysine binds in the vicinity of Lys39, Tyr113 and two IS1 residues (Cys38 and Cys99, Figure 5D). The glycated Lys262 in complex with the V-domain binds in the vicinity of Lys39 (located near IS1), Ala41 (IS1) and Pro42 (Figure 5E). Among the obtained conformations of the complexes of RAGE with Lys317- and Lys378-glycated HSA, there were none in which the CML would interact with IS1, IS2 or IS3. In the case of Lys317, it might be due to the intramolecular salt bridge between the CML and Lys313 of gHSA (Figure 5F). So, for the analysis and subsequent MD simulation, we chose the most probable (according to the ZDOCK SF) conformations from those in which there is no overlap of the albumin molecule with the C1-domain. In the selected complexes, modified Lys317 interacts sterically with Arg116, Ile120 and Pro121 (Figure 5F), while glycated Lys378 forms a salt bridge with Arg77 and interacts sterically with Val78 and Leu79 (Figure 5G). As mentioned above, in Lys525-glycated HSA, the CML forms an intramolecular salt bridge with the Arg521 and a hydrogen bond with the Gln522 of albumin. For this reason, in the complex with RAGE, the carboxyl group of the modified Lys525 does not form short-range electrostatic interactions with the receptor residues. In this complex, the CML binds in the vicinity of the Glu50 (belongs to IS2), Lys52 and Trp61 of RAGE (Figure 5H). And finally, in the complexes of RAGE with Lys573- and Lys574-glycated HSA, the CML residues do not interact with the surface of the receptor at all. Therefore, in these cases, for the analysis and subsequent MD simulation, we chose the most populated conformations from those in which there is no overlap of gHSA with the C1-domain (Figure 5I,J).

The values of the ZDOCK SF for the selected conformations of the gHSA–V complexes are presented in Table 2. A higher SF value means a more favorable conformation.

According to the values presented in Table 2, the V-domain forms the most energetically favorable complexes with gHSA modified at Lys137, Lys233, Lys262 and Lys317.

### 2.4. Interaction of gHSA with RAGE according to MD Simulation

The conformational stability of the complexes of gHSA with the V-domain of RAGE, as obtained by protein–protein docking, was tested by 100 ns MD simulation. The structures obtained by 100 ns MD are available in PDB format in the Appendix A. Figure 6 shows the general view of the gHSA–V complexes after 100 ns of MD simulation. According to the obtained complexes, RAGE interacts with domain IA of gHSA in the case of glycated Lys64 (Figure 6A), with domains IA and IB (Figure 6B) in the case of glycated Lys73, and with domain IB (Figure 6C) in the case of glycated Lys137. When HSA is glycated at Lys233, RAGE contacts the surface of domain IIA and, partially, of domain IA (Figure 6D), and in the case of glycated Lys262 and Lys317 it contacts only domain IIA (Figure 6E and Figure 6F, respectively). When HAS is glycated at Lys378, the V-domain of the RAGE interacts with HAS domains IIB and IIIA (Figure 6G). In the case of Lys525-, Lys573- and Lys574-glycated HAS, RAGE binds to domains IB and IIIB (Figure 6H, Figure 6I and Figure 6J, respectively). Moreover, when HAS is glycated at Lys525, the V-domain of the RAGE does not merely interact with the surface of gHSA but penetrates deep into the protein globule into the space between domains IB and IIIB (Figure 6H).

According to the qualitative visual inspection of the resulting complexes, the V-domain of RAGE binds most strongly to gHSA glycated at Lys233, Lys378, Lys525 and Lys574. Next, we attempted to quantify the efficiency of the gHSA–RAGE interaction. In computational experiments, it is customary to evaluate the efficiency of the interaction between a ligand and a receptor by the value of the free binding energy (ΔG). However, for macromolecular complexes, it is complicated to estimate the entropy component. Therefore, for protein–protein complexes, it seems more rational to estimate the binding strength from contacts between proteins—their number and type (polar or non-polar). The more atoms are involved in the formation of polar contacts between the proteins, the stronger and more specific the complex. Such an approach was applied by Sartore et al. to assess the effectiveness of the interaction between the spike protein (S-protein) of severe acute respiratory syndrome-related coronavirus 2 and angiotensin-converting enzyme 2 [24]. In the present study, we also applied this approach and assessed the efficiency of gHSA binding to RAGE by the number and type of the short-range (3.5 Å) contacts between the proteins. The results are presented in Table 3 and Figure 7.

According to the obtained results, CML interacts with RAGE in eight of the ten complexes studied.

In the case of glycated Lys64, the gHSA–V complex appears to be stable. CML forms salt bridges with the Lys52 and Arg98 (belongs to IS2) of RAGE throughout the entire simulation (Figure 7A). Other amino acids of IS2 (Arg48 and Lys107) also take part in the interaction with albumin (Table 3). There are seven salt bridges formed between the proteins, five of which persist for more than 70 ns of the simulation. It is interesting to note that the Arg48, Lys52, Trp61, Lys62 and Arg98 of RAGE, which take part in the interaction with gHSA glycated at Lys64 (Table 3), are found to be involved in the binding of the S100P protein dimer [14]. In the case of Lys73-glycated HSA, CML loses its contact with the Lys107 of IS2 during the simulation (Figure 5B) and moves closer to IS1. For the last 30 ns of the simulation, CML interacts with the Lys39 located near IS1 via a salt bridge (Figure 7B); Lys37, which is another amino acid residue of IS1, is also involved in the interaction with gHSA (Table 3). In total, the RAGE and Lys73-glycated HSA form one hydrogen bond and six salt bridges, three of which persist for at least 50% of the simulation time. In the case of Lys137-glycated albumin, the modified lysine keeps a salt bridge with the Arg48 of IS2 throughout the simulation (Figure 7C). Met102 and Asn105, which also belong to IS2, sterically interact with gHSA (Table 3). In this complex, there are seven specific interactions (five salt bridges and two hydrogen bonds) between the proteins; however, most of them exist only during the last period of the simulation.

In the case of Lys233-glycated HSA, CML maintains a hydrogen bond with the Tyr113 of RAGE throughout the simulation and also, during the simulation, moves closer to the Lys39 located near IS1 and forms a salt bridge with this residue for the last 80 ns (Figure 7D). Additionally, CML forms steric contacts with the Ser111 of RAGE. Other amino acids of the IS1 (Lys37, Lys43 and Asn81) and IS2 (Lys107) sites also take part in the interaction with gHSA glycated at Lys233 (Table 3). In total, in the complex of RAGE with Lys233-glycated HSA, more than a hundred atoms form short-range contacts between the proteins. Also, this complex has the largest number of specific interactions (ten salt bridges and four hydrogen bonds, most of which exist throughout almost the entire simulation), which makes it the leader in interaction efficiency. Notably, in this complex, the Asp256 and Asp259 of gHSA are involved in the interaction with RAGE. These amino acids are closely adjacent to site Sudlow I of albumin, which binds such endogenous and exogenous compounds as thyroxine, azido-thymidine, warfarin and indomethacin [25]. Moreover, Asp256 is known to be involved in the binding of the cation of cobalt [26]. It cannot be ruled out that the glycation of the Lys233 of HSA and subsequent binding of the protein to RAGE can significantly affect the interaction of these substances with albumin.

In the case of glycated Lys262, during the simulation, CML moves closer to the Lys39 located near IS1 and forms a salt bridge with it (Figure 7E). However, this contact only exists for the last 10 ns of the simulation. Other amino acids of IS1 (Lys43) and IS2 (Asn105, Gly106 and Lys107) also take part in the interaction with albumin (Table 3). Notably, almost the entire set of RAGE amino acids involved in the binding of Lys262-glycated gHSA is also involved in the interaction with the S100A6 protein (namely, Lys43, Asn105, Gly106, Lys107 and Glu108) [13].

According to the protein–protein docking, the modified Lys317 sterically interacts with the Arg116, Ile120 and Pro121 of the V-domain of RAGE, while its carboxyl group forms an internal salt bridge with the Lys313 of HSA and does not interact with any RAGE residues (Figure 5E). The conformation of this gHSA–V complex is unstable. Although the amino acids Lys39 (located near IS1) and Lys107 (IS2) contact with albumin after 100 ns of the simulation, CML itself does not interact with any amino acids of RAGE (Figure 7F). This complex has the lowest number of short-range contacts, which makes it the weakest among the studied ones (Table 3). Therefore, Lys317-glycated HSA probably does not interact with RAGE in vivo. The main reason for this would be the intramolecular salt bridge between Lys313 and the modified Lys317 of gHSA.

In the case of the glycation of Lys378, the carboxyl group of CML forms a salt bridge with the Arg77 of RAGE (Figure 7G). However, this interaction only exists for 20 ns of the simulation (the first and the last 10 ns). Arg77 does not belong to any of the ISs described by Xie et al. [10]. However, in the complex of RAGE with HSA glycated at Lys378, there are five residues of ISs involved in the interaction with gHSA; namely, Arg29 (IS3), Lys37 (IS1), Arg48 (IS2) and Asn81 (IS1) (Table 3, Figure 7G). The proteins form seven specific contacts (one π–π interaction, one hydrogen bond and five salt bridges), four of them persisting for 50 ns or longer. By the number of nonspecific contacts (steric interactions), this complex is one of the leaders.

As described in Section 2.3, the modified Lys525 binds near the Glu50, Lys52 and Trp61 of RAGE according to the macromolecular docking (Figure 5H). During the simulation, the conformation of the complex changes. In the final confirmation of the MD trajectory, Lys525 interacts with the Arg48 of IS2 (Figure 7H). However, the approach between these residues only occurs at the very last stage of the simulation; the intramolecular interactions between the modified Lys525 and the residues Arg521 and Gln522 of gHSA prevent the formation of a strong ionic bond Lys525–Arg48. Nevertheless, more than a hundred short-range contacts are formed in the complex of the V-domain with Lys525-glycated HSA (Table 3), which makes it one of the strongest among those studied. In addition to Arg48, other amino acids of ISs also take part in the interaction with albumin: Gly40 (IS1), Met102 (IS2), Asn105 (IS2), Gly106 (IS2) and Lys107 (IS2) (Table 3). The complex forms one hydrogen bond and eight salt bridges, most of which persist for more than 60 ns. Notably, the Glu119, Pro180 and Asp183 of gHSA, which are involved in the interaction with RAGE, are located in the vicinity of the ligand-binding site III of HSA. Site III binds such ligands as lidocaine, bilirubin and hemin [25]. The binding of RAGE to this region of HSA may affect the pharmacokinetics and metabolism of these compounds.

The interaction of RAGE with gHSA glycated at Lys573 is a special case. In the complex obtained by macromolecular docking, CML did not interact with any of the amino acids of the V-domain of RAGE (Figure 5H). However, during the MD simulation, CML moved closer to the Arg29 of IS3 (Figure 7I); the salt bridge between these residues exists for 30 ns of the simulation. Other residues of the ISs are not involved in the interaction with gHSA (Table 3), as there are actually no specific interactions between the molecules. The complex of RAGE with Lys574-glycated HSA is strong, forming more than a hundred short-range contacts between the proteins, including four hydrogen bonds and five salt bridges (Table 3). However, this complex is a nonspecific one, since the CML does not contact with any amino acids of RAGE (Figure 7J). Therefore, even if such an interaction does occur in vivo, it probably does not lead to further signal transmission.

Thus, according to the totality of the results of the macromolecular docking and MD simulation, the complex of RAGE with Lys233-glycated gHSA is the most efficient. The complex of RAGE with gHSA glycated at Lys64 also has remarkable specificity (strong and stable interaction of CML with the receptor and a significant number of specific contacts between the proteins). The cases of RAGE’s interactions with glycated Lys262 and Lys525 are next on the list; these complexes are characterized by many stable ionic interactions but weaker contact between the modified lysine and the receptor. The cases of gHSA glycation at Lys64 and Lys262 are notable for the fact that these gHSA–V complexes are most similar to the RAGE–S100P and RAGE–S100A6 complexes, respectively; the same amino acids of RAGE are involved in binding to gHSA and S100P (in the case of glycated Lys64) or S100A6 (in the case of glycated Lys262). Finally, the complex of the receptor with Lys378-glycated gHSA is remarkable for its low specificity though large number of steric interactions. The remaining complexes are characterized by both weak specificity and a smaller number of steric contacts. According to our results, specific RAGE complexes with Lys317- and Lys574-glycated at HSA are not formed.

Similar to the gHSA models described in Section 2.2, we examined how the binding with the V-domain of RAGE affects the structural characteristics of gHSA. We calculated the time dependence of both the Rg and SASA of gHSA in complex with RAGE, as well as the RMSF of the Cα atoms of gHSA. The plots are presented in Appendix A (Appendix A, respectively). According to the obtained results, the structure of gHSA is most dramatically affected by the binding of the V-domain to glycated Lys137 (Appendix A). We believe that this is due to the mobility of domains IB and IIIB relative to each other. A similar effect, although to a lesser extent, is observed in the case of RAGE binding to glycated Lys574 (Appendix A). In this complex, the V-domain interacts with domains IB and IIIB of gHSA (Figure 6J). Moreover, the RAGE surface also interacts with Lys137 (non-glycated in this case, Table 3). That is, the interaction of RAGE with the region near the Lys137 of gHSA has a significant effect on the structure of albumin. Of particular note is the RAGE complex with glycated Lys525 (Appendix A). The starting Rg value for this complex is higher than in the case of other lysines, since according to the results described in Section 2.2, the glycation of HSA at Lys525 leads to significant structural changes in the protein globule and an increase in the Rg value (Appendix A). However, after binding to RAGE, the structure of gHSA no longer changes and even becomes slightly denser (more compacted), partially compensating for the increase in the Rg after the glycation of Lys525. For the remaining lysines, the interaction of gHSA with RAGE does not significantly affect albumin’s compactness (Appendix A).

In the cases of the glycation of Lys137 and Lys574, as expected, the gHSA molecule is characterized by an increase in the SASA value (Appendix A). A similar effect, although to a lesser extent, occurs in the RAGE complex with gHSA glycated at Lys573 (Appendix A), in which the V-domain also interacts with domains IB and IIIB of gHSA (Figure 6I). This is also likely due to the movement of domains IB and IIIB relative to each other. It is also interesting to note the increase in the SASA in the case of glycated Lys378 (Appendix A). We believe this may be due to the fact that in this complex RAGE interacts with some amino acids of domain IIIA (namely, Gln385, Pro441, Glu442, Met446, Table 3), which can affect the conformation of domain IIIB (its susceptibility to external influences has been already mentioned in this section and in Section 2.2). For other lysines, the interaction of gHSA with RAGE does not significantly change the value of albumin’s SASA (Appendix A).

As for the RMSF, RAGE’s interaction with glycated Lys137 and Lys573 causes the greatest fluctuation in the HSA amino acids (Appendix A), with the maximum expectedly occurring in domains IB and IIIB. This effect is slightly less pronounced in the RAGE complexes with glycated Lys64 and Lys73 (Appendix A). For the remaining lysines, the fluctuations are even less pronounced (Appendix A). In general, comparing the structural changes in the gHSA molecule after glycation (Appendix A) and after RAGE binding (Appendix A), one can conclude that it is glycation itself, and not the interaction with RAGE, that has a greater effect on the characteristics of gHSA.

### 2.5. Interplay between Glycation and Binding of Fatty Acids to HSA.

Albumin is the main transport protein for the transfer of fatty acids (FAs) throughout the vascular bed [27]. Using X-ray analysis, it was shown that there are seven main FA binding sites in the albumin molecule: FA1 (domain IB), FA2 (domains IB and IIA), FA3 (domain IIIA), FA4 (domain IIIA), FA5 (domain IIIB), FA6 (domains IIA and IIB), and FA7 (domain IIA) [27]. According to the literature, FA5 is the highest-affinity FA-binding site [28]. The total concentration of major FAs in the blood of humans can reach 100 μM, that is, their concentration is comparable to the concentration of HSA. It is known that FA binding affects the interaction of albumin with the ligands of sites Sudlow I and II [29,30]. As part of our studies on the integrative role of albumin, we questioned whether FA binding could influence the glycation of albumin, and vice versa. Figure 8 shows the HSA molecule with seven molecules of oleic acid (OLA, one of four major FAs) bound in sites FA1–7 (PDB entry 1GNI [31]), and eight lysine residues highlighted that play a key role in the binding of gHSA to RAGE according to the results presented in the previous sections.

The proximity of site FA5, which has the maximum affinity for FAs, and Lys525, which is the main glycation site of albumin and plays an important role in the interaction with RAGE, deserves particular attention. The NH3 groups of the other studied lysines are not in close proximity to the OLA molecules, so we believe their glycation will have less of an effect on the binding of FAs at sites FA1–7 (however, it is possible that the lysine residues that we have not studied here may have such an effect). Thus, it is logical that the first step should be to evaluate how Lys525 glycation and OLA binding at site FA5 affect each other.

We performed molecular docking of β-D-glucopyranose (GLUC) near the Lys525 of HSA. Two models of albumin were used for docking: (1) free HSA and (2) HSA with an OLA molecule bound in site FA5 (Figure 9). The α- and β-anomers of glucose are in equilibrium, passing into each other through the acyclic form. The β-anomer was chosen due to the fact that the equilibrium is shifted toward this anomer. When glucose interacts with lysine, a nucleophilic attack occurs by the lysine side chain nitrogen atom (Nζ) on atom C1 of glucose (according to the standard numbering of the International Union of Pure and Applied Chemistry, IUPAC). In the HSA–GLUC complexes obtained by molecular docking, the glucose molecule binds near the amino acids Glu518, Arg521 and Glu556, which form a site different from site FA5. In the OLA-free HSA–GLUC complex, the distance between atom C1 of glucose and atom Nζ of Lys525 is 3.5 Å (Figure 9A), while in the HSA–GLUC complex with the OLA molecule it is 3.7 Å (Figure 9B). In both cases, the distance is sufficient for the nucleophilic attack of nitrogen Nζ on carbon C1. In both complexes, the estimated free energy of the binding of glucose to albumin (ΔG) is −3.8 kcal/mol. Based on the totality of all the above results, we believe that the OLA molecule bound in site FA5 apparently does not affect the possibility for glycation of Lys525.

On the other hand, the modified Lys525 turning into CML acquires a negatively charged group, which logically does not contribute to the binding of the OLA molecule, which also has a negatively charged group in its structure. To confirm this assumption, we applied the method of MD to study the conformational changes in native and Lys525-glycated HSA with an OLA molecule bound at site FA5. The conformation of the OLA molecule after 100 ns of MD simulation is shown in Figure 10 in 3D (left) and 2D (right) representations. According to the obtained results, in the case of both native HSA (Figure 10A) and gHSA (Figure 10B), the OLA molecule remains bound in site FA5 throughout the entire simulation. Its carboxyl group interacts with the hydroxyl group of Tyr401 and NH3 group of Lys525 (Figure 10A,B).

To study the effect of glycation on the interaction of OLA with site FA5 in more detail, we calculated the energy and conformational characteristics of the OLA–HSA and OLA–gHSA complexes from the obtained trajectories. The results are presented in Appendix A. The following parameters were calculated: the Rg and SASA of (g) HSA, as well as the RMSF of the Cα atoms of (g) HSA. Additionally, we calculated such parameters as the E_LJ-SR_ and E_COUL-SR_. These parameters are the energies of the van der Waals forces (Lennard–Jones potential, LJ) and Coulomb interactions, respectively, as calculated for the short-range contacts between the OLA and (g) HSA atoms. As seen in Appendix A, the glycation of Lys525 has little effect on the Rg, SASA and E_LJ-SR_. The average values of the E_LJ-SR_ over 100 ns of the simulation are −154 and −150 kJ/mol for native HSA and gHSA, respectively. As for the RMSF, as expected, the glycation of Lys525 leads to the slightly greater conformational mobility of domain IB compared to the native protein due to the fact that domains IB and IIIB (to which Lys525 belongs) are spatially close together (Figure 8). The most significant difference was found for the E_COUL-SR_ value. The average E_COUL-SR_ value over 100 ns of simulation is −185 kJ/mol for native HSA and −56 kJ/mol for gHSA. This difference is apparently achieved due to the additional negatively charged carboxyl group of CML located in proximity to the carboxyl group of OLA. The plot of the E_COUL-SR_ for gHSA (Appendix A) highlights a region from 45 to 60 ns, when the electrostatic energy is close to zero. During this period of the simulation, the hydroxyl group of Tyr401 moves away from the OLA and does not form close contacts with its carboxyl group, thus weakening the electrostatic interaction.

Thus, according to the results of the molecular modeling, the glycation of Lys525 reduces the affinity of site FA5 for OLA (and, highly likely, for other FAs). Electrostatic interactions play a key role in this effect.

### 2.6. Probing of HSA–RAGE Dimers’ Structure

At the next stage, on the basis of the gHSA–V complexes described in Section 2.4 (Figure 6), in a pilot experiment we attempted to construct the possible structures of the RAGE–gHSA dimers. As noted in Section 2.4, specific RAGE complexes with gHSA glycated at Lys317 and Lys574 were not formed. Therefore, we excluded Lys317 and Lys574 from the list of candidate monomers for the formation of dimers. The complex of the V-domain with Lys73-glycated gHSA, according to the MD simulation, takes the conformation in which albumin contacts the “forbidden” region of RAGE (the region of contact of the V- and C1-domains of the receptor). We will study the case of Lys73 further in the future: we will need to include the C1-domain into the model of RAGE for the computational experiment. In the meantime, at the current stage, we also excluded Lys73 and did not consider the complex of RAGE with Lys73-glycated gHSA when constructing the dimers. Thus, complexes of the V-domain with gHSA glycated at Lys64, Lys137, Lys233, Lys262, Lys378, Lys525 and Lys573 were used to construct the dimers. For this purpose, we performed a pairwise alignment of the gHSA–V complexes with the structure of the VC1C2-domains of RAGE obtained experimentally (PDB entry 4LP5, chain A [12]). The alignment diagram is shown in Figure 11.

As mentioned above, Xue et al. proposed a model of the RAGE complex with the S100 protein: S100 is sandwiched between two RAGE monomers, which contact with another RAGE monomer not involved in S100 binding [16]. Moreover, in the RAGE dimer that interacts with S100, the distance between the monomers of the C2-domain exceeds 100 Å.

Table 4 shows the distances between the monomers of the C2-domain of RAGE in the complexes of the RAGE dimers with gHSA that we constructed. An asterisk indicates the complexes in which the V-domain monomers are superimposed on each other, which means that such a complex cannot be formed.

A total of 21 complexes were built, and some of them–the most typical—are shown in Figure 12 as an example. The dimers marked with an asterisk in Table 4 cannot be formed in vivo because the V-domains overlap in these complexes when aligned. An example of such a dimer is shown in Figure 12E; the overlap region is marked in red.

In the RAGE molecule, the region of the polypeptide chain connecting the C1- and C2-domains is quite flexible, so all the constructed dimers in which there is no overlap of the V-domains can potentially be formed under real conditions (note that the C2-domains of RAGE are adjacent to the plasma membrane; Figure 1 and Figure 2). However, we think that the most promising models of the dimers are those that can be docked onto the surface of the plasma membrane without additional conformational changes. These are the dimers between the following pairs of lysines: Lys64–Lys233 (Figure 12B), Lys64–Lys262, Lys137–Lys378, Lys137–Lys525 (Figure 12D), Lys262–Lys378 (Figure 12F). Since the complexes of the V-domain with gHSA glycated at Lys233, Lys64, Lys525, Lys262 and Lys378 were found to be the strongest (see Section 2.4), we consider the Lys64–Lys233, Lys64–Lys262 and Lys262–Lys378 dimers to be the leading ones. The structures of these dimers optimized by energy minimization using GROMACS 2019.4 software (University of Groningen, the Netherlands) [23] are available in PDB format in the Appendix A. In future studies, we plan to check the conformational stability of the selected dimers using MD simulation.

In Section 2.1 and in Figure 2, we schematically described the possible structures of the dimers of RAGE in complex with different macromolecules. According to the results described in the current section, in the case of gHSA, the scheme presented in Figure 2A is most likely implemented: the RAGE dimer with albumin has a similar structure to the RAGE–S100A6 and RAGE–S100P dimers.

### 2.7. Comparative Analysis of Primary Sequences of Albumins from Various Species

Using the Clustal Omega online service [32], we performed a detailed comparative analysis of the primary sequences of albumin from different species to establish if there are interspecific differences in the molecular characteristics of albumin that determine its interaction with RAGE. Figure 13 shows the alignment of the primary sequences of albumin from the main representatives of various mammalian species: HSA, BSA, rat (RSA), equine (ESA), and leporine (LSA) serum albumin. Additionally, we have included chicken albumin (CSA) in the list for comparison. The primary albumin sequences from the UniProt database [33] were used for analysis (entries P02768, P02769, P02770, P35747, P49065 and P19121 for HSA, BSA, RSA, ESA, LSA and CSA, respectively).

Next, we analyzed how the HSA amino acids involved in the interaction with RAGE are conserved. For the analysis, we selected the strongest gHSA–RAGE complexes according to the results of the molecular modeling; namely, RAGE complexes with HSA glycated at Lys233, Lys64, Lys262, Lys525, and Lys378 (in descending order of efficiency of the interaction between gHSA and RAGE). The results are presented in Table 5. It is interesting to note that, in the case of glycation at Lys233, all the albumin amino acids that interact with RAGE are highly conserved across the mammals (only Glu208 and Glu230 are replaced by Asp in LSA; Figure 13). When CSA is added to the list for comparison, conservation is reduced as expected, although not critically: only two amino acids undergo non-homologous substitutions in CSA: Thr236 changes to His, and Asp259 to Arg. A similar situation can be observed in the case of glycated Lys64. In the case of Lys262 glycation, the amino acids interacting with RAGE are conserved among mammals, but in CSA, they undergo greater changes compared to the cases of Lys233 and Lys64 (Figure 13). Regarding glycated Lys525 and Lys378, the HSA amino acids interacting with RAGE are the least conserved of all the complexes examined.

We also analyzed the number of lysines in these albumins and determined how many of them lie on the surface of the protein and are available for interaction with the receptor. The PDB structures 3JQZ [34], 6QS9 [35], 6XK0 [36] and 8BSG [37] were used as three-dimensional models of HSA, BSA, ESA and LSA, respectively. The three-dimensional model of RSA that we previously built using the homology modeling method [38] and a three-dimensional model of CSA from the AlphaFold database [39] were also used. The results are presented in Table 6.

Among mammals, there is a tendency that the more omnivorous the animal, the fewer lysines there are on the surface of its albumin. Regarding CSA, it is interesting to note that the glucose concentration in the blood plasma of birds is 50–100% higher than in mammals of a similar weight; this feature is called benign hyperglycemia. However, chicken albumin is glycated to a lesser extent than bovine albumin, even when the two albumins are exposed to increasing concentrations of glucose up to 500 mM in vitro [40]. Benign hyperglycemia is a common physiological feature of birds, and the development of mechanisms of resistance to albumin glycation appears to be inextricably linked to their evolution. Comparative analysis of the reconstructed albumin sequences indicates that the ancestor of birds had 6–8 fewer lysine residues in the albumin molecule compared to mammalian albumin [40]. The development of benign hyperglycemia in birds is believed to have coincided with a radical genomic rearrangement that resulted in the loss of important genes. Among these genes, there is the gene of glucose transporter type 4, which is a transporter responsible for the insulin-dependent transport of glucose in insulin-sensitive cells of other vertebrates. This loss appears to have resulted in the re-organization of the insulin-dependent signaling pathway in avian tissues [41].

## 3. Discussion

First, we should discuss the limitations of the presented computational experiments, which are mainly connected with the accuracy of the protein–protein docking procedure. Here, we performed macromolecular docking with the help of the ZDOCK 3.0.2 software (UMass Chan Medical School, MA, USA, https://zdock.umassmed.edu/, accessed on 23 January 2024) [42]. The developers of the program regularly report the results of critical assessment of the predicted interactions (CAPRI reports) [43,44,45]. According to the tests, the protein–protein complexes might be predicted incorrectly [45]. Thus, concerning the complexes of gHSA with the V-domain obtained by us, we can only claim that these are some of the possible conformations of the complexes. From the array of the obtained configurations of gHSA–V, we selected the most favorable ones based on the ZDOCK SF. Moreover, an extra selection criterion was the experimental data of Xie et al. concerning the regions on the surface of the V-domain that interact with AGE peptides and full-length gBSA [10]. The conformational stability of the complexes was tested by MD simulation. However, it is possible that the “true” gHSA–RAGE conformation is not among the most favorable ones according to the ZDOCK SF. In the future (not until experimental data are obtained), it seems reasonable to check the less probable conformations among those where CML interacts with ISs, as well as to apply other programs for macromolecular docking to check the reproducibility of the obtained conformations.

Among the limitations of the presented work, it should also be mentioned that we considered the glycation of only 10 lysine residues out of 59 lysines of HSA and studied only 1 model of modification (carboxymethyl-lysines). So far, the mechanism of interaction of other models of gHSA (carboxymethyl-arginines, glyoxal-lysine-amides, carboxyethyl-lysines, etc.) with RAGE, as well as the effect of gHSA oxidation on its dimerization and interaction with RAGE, remain unexplored. These are tasks for our future research.

The first papers demonstrating that gHSA can bind to the RAGE of endothelial cells were published in the 1990s [7,8]; much later, data appeared on the binding of glycated albumin with the RAGE of adipocytes [9]. Since then, evidence has accumulated on the major role of the interaction of gHSA with RAGE in the pathogenesis of DM and other diseases [46]. For example, in obesity, increased levels of AGEs, mainly gHSA, fuel both oxidative stress and the AGE/RAGE axis, which in turn can increase inflammation in already inflammatory tissue, thereby accelerating disease progression [47]. One of the pathways of kidney damage in DM and the development of diabetic nephropathy is the transdifferentiation of renal tubular cells into myofibroblasts, which occurs when RAGE is activated. These processes induce the generation of transforming growth factor beta and other cytokines that mediate the transdifferentiation [48].

Thus, if a means could be found to suppress the gHSA–RAGE interaction, this could provide a method of inhibiting cellular activation and restricting long-term tissue destruction in chronic pathologies. To achieve this aim, the first task is to describe the molecular mechanism of gHSA interaction with the receptor. However, there are still not many studies devoted to this problem. Xie et al. used the NMR technique to study the interaction of AGE peptides and glycated full-length BSA with RAGE and found that only the V-domain of the receptor participates in binding [10]. The authors also identified the residues of the V-domain that participate in the interaction with AGE peptides and gBSA. Moreover, Xie et al. established that the affinity of gBSA for the monomer of V-domain is lower than for the full-length receptor, and they assumed that in vivo dimerization or even oligomerization of the receptor occurs when albumin binds.

Despite the significant contribution to the study of the mechanism of gBSA binding to RAGE, Xie et al. did not consider which modified lysines of BSA participate in the interaction with the receptor [10]. Such an attempt was made by Tramarin et al. [49]. According to the data obtained by the authors (combining in vitro and computational experiments with HSA), the modified Lys525 of HSA and Arg98 of RAGE play the major role in the interaction of these protein molecules (here and below, we provide RAGE residue numbers according to the UniProt database [33], entry Q15109, while Tramarin et al. used numbering that does not include the signal peptide, and the Arg98 mentioned here corresponds to Arg78 in [49]). Also, according to Tramarin et al., the Glu184, Glu188, Glu400, Glu518, Glu556 and Lys432 of gHSA, as well as the Arg114, Glu108, Lys107 and Lys110 of RAGE, also take part in the interaction of albumin with the receptor [49]. This is partially consistent with our results: according to our calculations, the interaction of RAGE with Lys525-glycated HSA involves the Lys107 and Glu108 of the receptor, as well as the Glu184 and Glu518 of albumin. According to our results, other residues mentioned in [49] also participate in the interaction with gHSA, however, with gHSA glycated not at Lys525 but at other lysines. Thus, the Arg98 and Lys107 of RAGE participate in the interaction of RAGE with Lys64-glycated HSA; Arg114—with Lys73-glycated HSA; Glu108—with Lys137-glycated HSA; Lys107, Glu108 and Lys110—with Lys233-glycated HSA; Lys107 and Glu108—with Lys262-glycated HSA (Table 3). It is possible that the partial contradiction may be connected with the fact that Tramarin et al. studied another type of albumin glycation, namely Amadori adducts, which precede the formation of AGEs. However, both the work of Tramarin et al. [49] and our computational experiments have shown that the albumin domains IB and IIIB participate in the interaction of Lys525-glycated HSA with the receptor. It is also important to note that the Glu518 and Glu556 of gHSA, which according to Tramarin et al. surround the Amadori adduct of Lys525 and are involved in the interaction with RAGE, according to our results take part in the binding of glucose by native albumin.

Earlier, the same scientific group performed macromolecular docking of the V-domain with HSA, modified at Arg472 (N-2-pyrimidyl-ornithine), Lys436 (dihydropyridine-lysine) and Lys262 (N-(3-formyl-3, 4-dehydro-piperidinyl) lysine) [50]. We did not study modified arginines, and Lys462 did not come into our field of view since it is not located on the surface of HSA (dihydropyridine-lysine has a longer side chain compared to CML and therefore has more chances to reach the protein surface and become available for interaction with RAGE). As for Lys262, in the research by Mol et al. [50], the amino acids involved in the interaction with the modified Lys262 do not match those involved in this interaction, according to our results. We suggest two reasons for this discrepancy: either the type of lysine modification matters, or different conformations of HSA–RAGE complexes are possible even when the same residue is modified. Since albumin contains 59 lysines, 24 arginines and 16 histidines, the number of their possible modifications and interactions with RAGE is enormous.

In biochemical experiments in vivo, rodents are the main species of experimental animals, and this is true for studies related to metabolic disorders in the human body. In our numerous experiments, we have shown that HSA is closer to RSA than to BSA based on its biochemical properties [51,52]. Therefore, RSA and HSA are interchangeable in the experiments devoted to studying the role and characteristics of albumin. However, in experiments with albumin, authors often use BSA due to its low cost. In particular, gBSA is used as a source of AGEs in studies of rodent or human cell cultures [53,54].

At the next stage, we compared the structures of the gHSA–V complexes obtained here with structure 2L7U from the PDB database [55]. Structure 2L7U is the experimentally resolved three-dimensional structure of the complex of the V-domain of human RAGE with the fragment of BSA Asp124–Glu125–Phe126–CEL127–Ala128–Asp129–Glu130 (BSA^124−130^) [55], where CEL is carboxyethyl-lysine. In 2L7U, CEL interacts with the RAGE residues Arg98, Gln100 and Lys110 (according to the UniProt numbering). In HSA, peptide BSA^124−130^ corresponds to peptide Thr125–Ala126–Phe127–His128–Asp129–Asn130–Glu131; that is, in the corresponding HSA peptide, lysine is replaced by histidine. Histidine residues can also be glycated, and in one of our next works, we will definitely consider the interaction of His128-glycated HSA with the V-domain of RAGE. It can be noted that, among the 10 gHSA-V complexes obtained by us, the complex of RAGE with Lys64-glycated HSA is the closest to structure 2L7U. In both structures, the glycated lysines (CML64 in gHSA and CEL127 in BSA^124−130^) interact with Arg98. However, when the structures are superimposed, the position of CML relative to RAGE in the gHSA–V complex obtained in silico does not coincide with the position of CEL in the complex BSA^124−130^–RAGE obtained by NMR [55] (Figure 14A). We believe that this is not a result of the limitations of our computational experiments. The short peptide consisting of 7 amino acids is not the same as the full-length albumin molecule consisting of more than 580 residues. The result of the alignment of structure 2L7U (complex BSA^124−130^–RAGE, [55]) with the full-length BSA molecule (PDB entry 6QS9, chain A [35]) and with the structure of VC1-domains of RAGE (PDB entry 4OI8, chain A [15]) is presented in Figure 14B. When these structures are superimposed, the C1-domain of the receptor significantly overlaps with the BSA molecule (overlapping area is outline in red in Figure 14B); that is, the conformation of the complex of RAGE with Lys127-glycated full-length BSA must differ from the conformation of the complex of RAGE with AGE-peptide BSA^124−130^.

Therefore, the use of gBSA in studies of the molecular mechanisms of human albumin’s involvement in the AGE/RAGE axis can only be pursued with the understanding that there are interspecies differences that may affect the molecular mechanism of the gSA–RAGE interaction. According to our results, the main type of AGE–HSA that interacts with human RAGE is HSA glycated at the conserved Lys233 residue. However, in the albumin of other mammals, additional glycation sites that interact with RAGE may appear (Table 6). Therefore, it is reasonable that the present study performed with HSA should be repeated with BSA and RSA. Whilst horse, rabbit and other farm animals are much lesser objects of research in the field of metabolic disorders, information on the mechanism of albumin interaction with the AGE/RAGE axis in these species could be useful, for example, for the selection of feed or in veterinary medicine.

As mentioned, albumin is the main transporter of FAs in the vascular bed, and under in vivo conditions, some of the protein molecules are loaded with FAs. According to our assessment, the binding of FAs to HSA does not significantly affect the possibility of protein glycation, at least not in the case of major glycation sites. On the other hand, according to our results, the glycation of albumin weakens its interaction with FAs. The same result is also obtained in a number of experiments. Thus, albumin extracted from the plasma of patients with type 2 DM was less efficient in binding non-esterified fatty acids compared to a control group of healthy subjects [56]. The glycation of albumin under in vitro conditions reduced its affinity for α-parinar [57], oleic, linoleic, lauric, caproic [58] and stearic [59] acids. With an increased level of free FAs not bound to albumin, the mode of their interaction with receptors changes, which can become one of the pathogenetic factors of non-alcoholic steatohepatitis and metabolic syndrome [60,61,62].

Another modification of albumin that frequently occurs in vivo and which we did not cover in our work is its oxidation. The HSA molecule contains a free thiol group within Cys34, which can serve as a trap for reactive oxygen species [63]. In healthy people, about 30% of the albumin is oxidized [64], whereas under pathological conditions accompanied by oxidative stress, the level of oxidized albumin can increase up to 70% [65]. Data on the effect of glycation on the antioxidant properties of albumin are contradictory: in some cases, the antioxidant properties are weakened, while in others, they are enhanced [66,67,68,69]. It has been suggested in the literature that the reasons for the inconsistent behavior of glycated albumin in vitro may be interspecies differences, the nature and concentration of the carbohydrates involved (glucose, methylglyoxal), and the applied incubation conditions with monosaccharides [70]. With the help of MD simulation, Jeevanandam et al. showed that the glycation of some HSA arginines to methylglyoxal-hydroimidazolone changes the secondary structure of the protein in domain IA, which leads to the distancing of Cys34 and Tyr84 from each other and, as a consequence, to the increased accessibility and reactivity of the thiol group of Cys34 [71].

The combined effects of glycation and oxidation may accelerate the development of comorbidities in DM. While glucose itself contains a carbonyl group that is involved in the initial glycation reaction, the most important and reactive carbonyls are formed under oxidative stress and damage either carbohydrates (including glucose itself) or lipids. The resulting carbonyl-containing intermediates then modify proteins, yielding “glycoxidation” and “lipoxidation” products, respectively. This common pathway of glucose- and lipid-mediated stress is the basis of the carbonyl stress hypothesis [72]. How the co-glycation and oxidation of HSA affect its interaction with RAGE remains to be elucidated. This is especially important considering that with an increase in the reactivity of the Cys34 thiol group of albumin, the probability of the formation of albumin dimers through the S–S bridge increases. In the previous sections, we described the experimentally obtained complexes of the V-domain dimer with the dimers of the proteins S100P (PDB entry 2MJW [14]) and S100A6 (PDB entry 2M1K [13]). In these complexes, due to the dimerization of S100, symmetry is maintained: the same amino acids of the V-domain monomers interact with the S100 dimer. The albumin molecule is asymmetric, so the complexes of the RAGE dimers with gHSA that we constructed are also asymmetric and not all of them look natural (see, for example, Figure 12C). A different picture may emerge after HSA dimerization: in this case, the formation a symmetrical gHSA–RAGE dimer becomes possible.

One more point of practical significance for the obtained results can be noted. HSA is the major transport protein for many endogenous and exogenous ligands; the protein molecule contains three major and several minor ligand-binding centers. Moreover, albumin is liable to undergo allosteric modulation (binding of a ligand in one site changing the binding activity of other sites) [73]. It is known that glycation reduces the affinity of albumin for ligands of Sudlow sites I and II [74] and for FAs [56,57,58,59]. Whether the binding of gHSA to RAGE would alter its binding activity toward FAs and the Sudlow sites’ ligands remains unclear. On the other hand, the susceptibility of HSA to allosteric modulation may be a key to the management of its interaction with RAGE.

## 4. Materials and Methods

### 4.1. Preparation of the Three-Dimensional Models of Ligands and Proteins

Three-dimensional models of low-molecular ligands were built and optimized with the help of Avogadro v1.2.0 software (©2022 Avogadro Chemistry, University of Pittsburgh, Pittsburgh, PA, USA) [75]. The crystal structure of HSA (PDB ID 3JQZ, chain A [34]) was taken to prepare the three-dimensional model of the protein. The structure of the RAGE V-domain monomer obtained by NMR (PDB ID 2M1K, chain A [13]) was selected to prepare the three-dimensional model of the V-domain (more details about the choice of the models of RAGE are provided in Section 2.1). The expression tag of 2M1K (Ala21 and Met22) was retained. Unused chains, ligands and water molecules were deleted from the 3JQZ and 2M1K structures, and the missing atoms were built with the help of the Visual Molecular Dynamics v.1.9.4a53 software package (VMD, University of Illinois Urbana-Champaign, IL, USA) [76].

### 4.2. Building of the gHSA Model

Carboxymethyl-lysine (CML) was chosen as the model of gHSA. The models of the modified lysines were built as follows. The topology of CML was described based on the available information on the atomic charges, bond lengths, bond and torsion angles for different types of atoms and atomic groups presented in the CHARMM27 force field [77] topology files of the GROMACS 2019.4 software (University of Groningen, the Netherlands) [23]. The protonation states of the CML carboxyl group (deprotonated) and side chain nitrogen (double protonated) were assigned according to the physiological pH values. The prepared topology has been added to the library of GROMACS 2019.4 [23]. Then, molecular docking of the carboxymethyl group into the sites of glycation of HSA (in the vicinity of the selected lysines) was performed (more details about choice of modification sites are provided in Section 2.2). The details of the molecular docking method are described below in Section 4.3. Based on the topologies added into the library, the models of gHSA with glycated lysines were generated using the GROMACS 2019.4 software [23]. The resulting gHSA structures were optimized by MD simulation. The details of the MD method are described below in Section 4.5.

### 4.3. Molecular Docking

Molecular docking of low-molecular-weight ligands’ HSA sites was performed using the Autodock Vina 1.1.2 software (The Scripps Research Institute, La Jolla, CA, USA) [78]. A search area of 15 × 15 × 15 Å3 was set in the studied protein binding site. The parameter “exhaustiveness” (determining the number of runs and the amount of computational effort) was set to 10. The parameter “energy_range” (maximum energy difference between the best binding mode and the worst one) was set to 3 kcal/mol. The number of the most optimal conformations in the output file (num_modes) was set to 10. For further proceeding, we selected those conformations of the complexes in which the distance between the functionally significant atoms was minimal. In the case of the docking of the carboxymethyl group, this was the distance between the nitrogen atom of the side chain of the lysines and the saturated carbon atom of the carboxymethyl group; in the case of the docking of GLUC, this was the distance between carbon atom C1 of GLUC (according to the standard IUPAC numbering) and atom Nζ of Lys525.

### 4.4. Macromolecular (Protein–Protein) Docking

The prepared models of gHSA and the V-domain of RAGE were used for macromolecular docking. Macromolecular docking was performed with the help of the online version of the ZDOCK 3.0.2 software (UMass Chan Medical School, Worcester, MA, USA, https://zdock.umassmed.edu/, accessed 23 January 2024) [42]. The glycated lysine was assigned as the binding site of gHSA. For the V-domain, the entire surface of the protein was assigned as the search area. In the ZDOCK 3.0.2 software (UMass Chan Medical School, Worcester, MA, USA, https://zdock.umassmed.edu/, accessed 23 January 2024), all the obtained conformations of the protein–protein complexes are ranked according to the value of the SF, which represents the energy characteristics of the complex and evaluates the probability of it being in a given conformation. The SF of the ZDOCK 3.0.2 software (UMass Chan Medical School, Worcester, MA, USA, https://zdock.umassmed.edu/, accessed 23 January 2024) includes the shape complementarity, electrostatics and a pairwise atomic statistical potential developed using the contact propensities of transient protein complexes [79]. The higher the value of the SF, the higher the probability of finding the complex in this particular conformation. The stability of the resulting complexes was tested by MD simulation. The MD procedure is described in more detail below in Section 4.5.

### 4.5. Molecular Dynamics

The conformational behavior of gHSA, gHSA complexes with RAGE, and HSA/gHSA complexes with OLA was studied by MD simulation using the GROMACS 2019.4 software [23] in the CHARMM27 force field [77]. The gHSA molecule (or gHSA–V and gHSA–OLA complexes) were virtually placed in a cubic periodic box filled with water molecules. The TIP3P water models (transferable intermolecular potential with 3 points) were used to describe the water molecules [80]. To neutralize the systems, sodium ions were added. The temperature (300 K) and pressure (1 bar) were kept constant using the V-rescale thermostat [81] and Parrinello–Rahman barostat [82], with coupling constants of 0.1 ps and 2.0 ps, respectively. The long-range electrostatic interactions were treated by the particle-mesh Ewald method [83]. The Lennard–Jones interactions were calculated with a cutoff of 1.0 nm. The LINCS algorithm (linear constraint solver for molecular simulations) was used to constrain the bonds’ length [84]. Before running the MD simulations, all the structures were minimized by steepest descent energy minimization and equilibrated under NVT (1000 ps) and NPT (5000 ps) ensembles. The time step for the MD simulation was 0.002 ps. The length of the simulation was 50 ns in the case of single gHSA molecules and 100 ns in the case of gHSA complexes with the V-domain and HSA/gHSA complexes with OLA. Two-dimensional diagrams of the ligand–protein complexes were prepared using LigPlot+ v.2.2 (European Bioinformatics Institute, Hinxton, Cambridgeshire, UK) [85].

## 5. Conclusions

In the present study, we explored the mechanism of interaction of albumin, glycated at certain lysine residues, with RAGE; described the main and minor interaction sites; and assessed the possibility of the mutual influence of HSA glycation and the albumin binding of fatty acids. We revealed that the binding of fatty acids to HSA does not affect the ability of Lys525 (the most reactive lysine) to be glycated. In contrast, modification of Lys525 reduces the affinity of albumin for FAs due to the energetically unfavorable proximity of the carboxyl groups of FAs and CML. A pilot computational experiment on the construction of the RAGE dimers with gHSA showed that the dimer is most likely formed by the same mechanism as the dimer of RAGE with the S100 proteins. Additionally, we examined how glycation and RAGE binding affect the structural features of gHSA, which is especially important given the fact that albumin—being the main transport protein of the body—has a significant influence on the pharmacokinetics of many drugs and the toxicokinetics of many toxics. Interspecies differences in the molecular structure of albumin have been identified, which may affect the mechanism of interaction of glycated albumin with the AGE/RAGE axis and affect the results of in vitro and in vivo experiments with glycated albumin as a source of AGE. The obtained information on the binding of gHSA to RAGE could be useful for understanding the role of gHSA in the pathophysiology of metabolic diseases and improve the methodology for studies of the cellular signaling pathways involving RAGE.

## Figures and Tables

**Figure 1 ijms-25-03204-f001:**
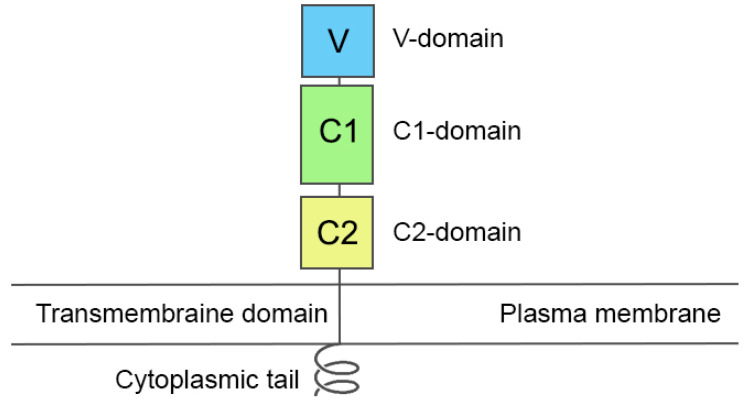
Structural organization of a monomer of the receptor for advanced glycation end products (RAGE). The extracellular domain consists of three domains: V, C1 and C2. The extracellular domain is connected to the intracellular tail by a single-stranded transmembrane domain.

**Figure 2 ijms-25-03204-f002:**
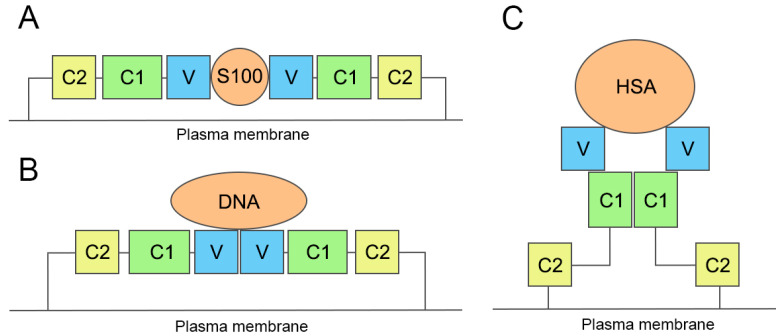
Schematic representation of the RAGE complexes with macromolecular ligands. (**A**) The complex of the RAGE dimer with S100, compiled on the basis of the experimental structure of the V-domain dimer in complex with S100A6 (PDB entry 2M1K [13]) and its alignment with the experimental structure of the full-length human RAGE extracellular domain (PDB entry 4LP5 [12]). (**B**) The complex of the RAGE dimer with DNA, compiled on the basis of the experimental structure of the dimer of the VC1-domains in complex with DNA (PDB entry 4OI8 [15]) and its alignment with the experimental structure of the full-length human RAGE extracellular domain (PDB entry 4LP5 [12]). (**C**) The complex of the RAGE dimer with human albumin (HSA), compiled on the basis of the structure of the RAGE–HSA oligomer proposed by Xie et al. [10].

**Figure 3 ijms-25-03204-f003:**
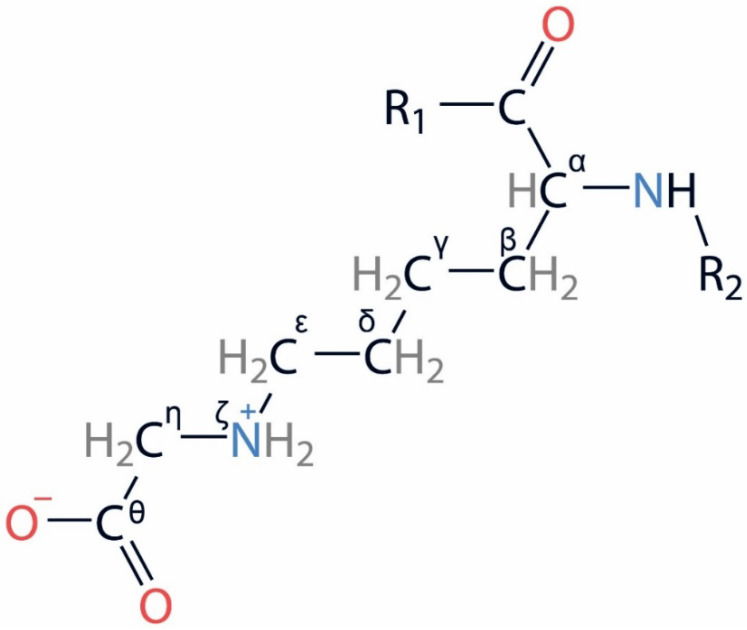
Chemical structure of the carboxymethyl-lysine (CML) within a polypeptide chain. The protonation state of the carboxyl group (deprotonated) and amine (double protonated) corresponds to physiological pH values. The Greek letters indicate the standard numbering of the atoms. The carbon, hydrogen, oxygen and nitrogen atoms are marked in black, grey, red and blue, respectively.

**Figure 4 ijms-25-03204-f004:**
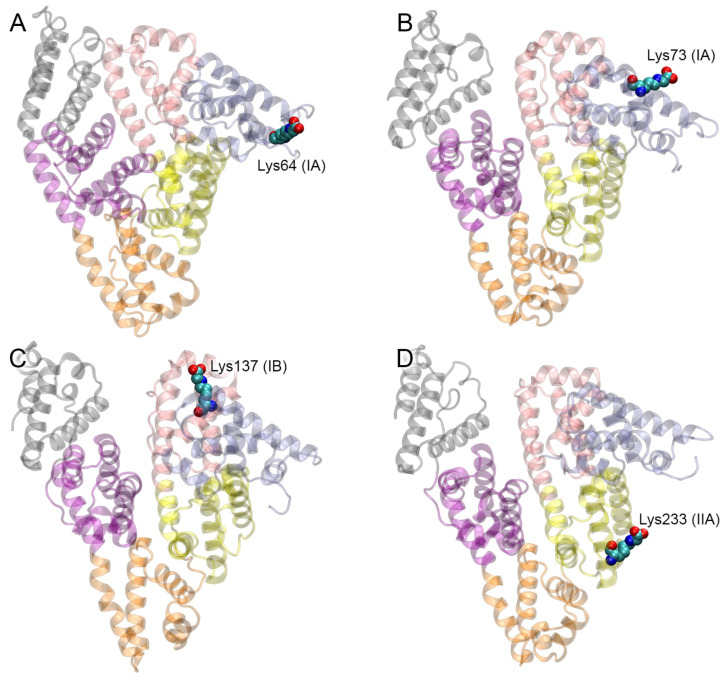
Three-dimensional structures of glycated human serum albumin (gHSA) modified at Lys64 (**A**), Lys73 (**B**), Lys137 (**C**), Lys233 (**D**), Lys262 (**E**), Lys317 (**F**), Lys378 (**G**), Lys525 (**H**), Lys573 (**I**) and Lys574 (**J**) obtained by molecular modeling. Glycated lysines are represented by colored spheres. The carbon, oxygen, and nitrogen atoms of glycated lysines are shown in cyan, red, and blue, respectively. Domains IA, IB, IIA, IIB, IIIA, and IIIB of gHSA are shown with ice blue, pink, yellow, orange, purple, and black ribbon, respectively. After the glycated lysine number, the gHSA domain to which it belongs is indicated in parentheses.

**Figure 5 ijms-25-03204-f005:**
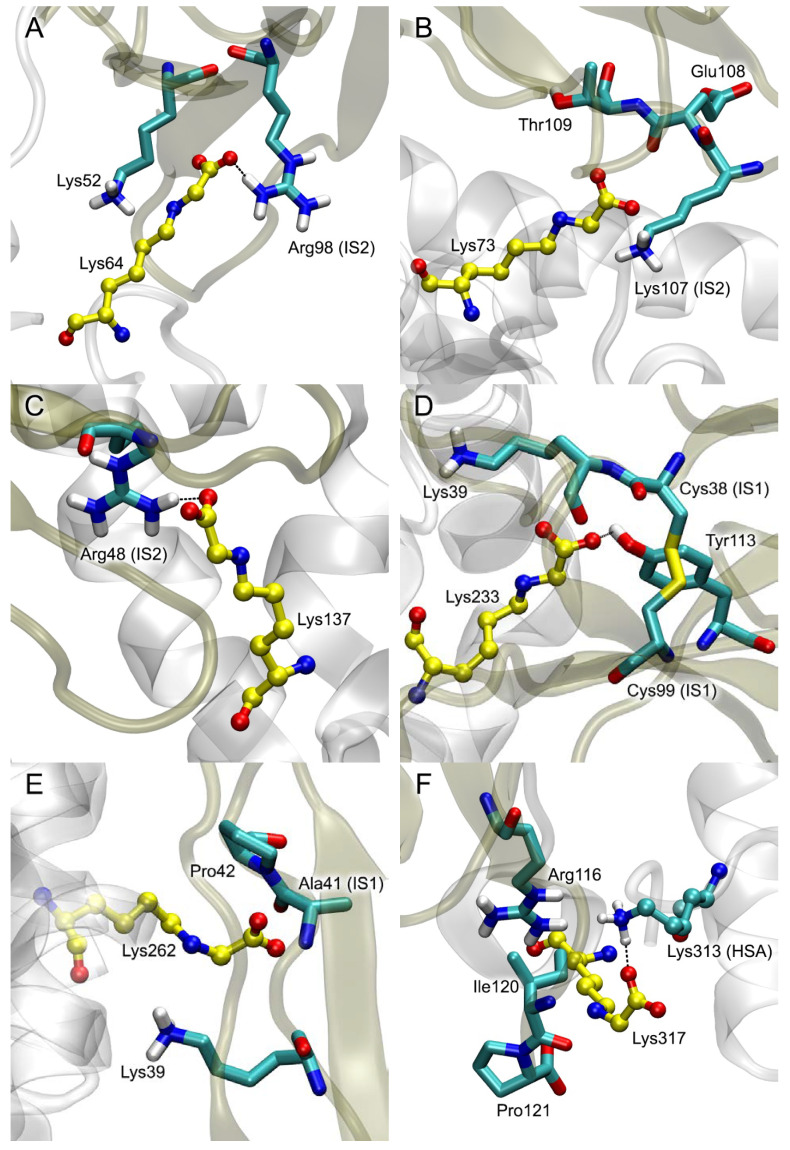
Macromolecular docking of gHSA modified at Lys64 (**A**), Lys73 (**B**), Lys137 (**C**), Lys233 (**D**), Lys262 (**E**), Lys317 (**F**), Lys378 (**G**), Lys525 (**H**), Lys573 (**I**) and Lys574 (**J**) to the V-domain of RAGE. The gHSA residues are shown in “ball and sticks” representations; the RAGE amino acids are shown in “sticks” representations. Carbon, hydrogen, oxygen and nitrogen atoms of amino acids are shown in cyan, white, red and blue, respectively. The carbon atoms of CML are highlighted in yellow. The gHSA backbone is shown with a gray ribbon; the backbone of the V-domain is shown with a brown ribbon. Non-essential hydrogens are omitted for clarity. IS1 and IS2 are interaction surfaces 1 and 2, which are the regions on the surface of RAGE that bind AGE peptides and gBSA according to Xie et al. [10].

**Figure 6 ijms-25-03204-f006:**
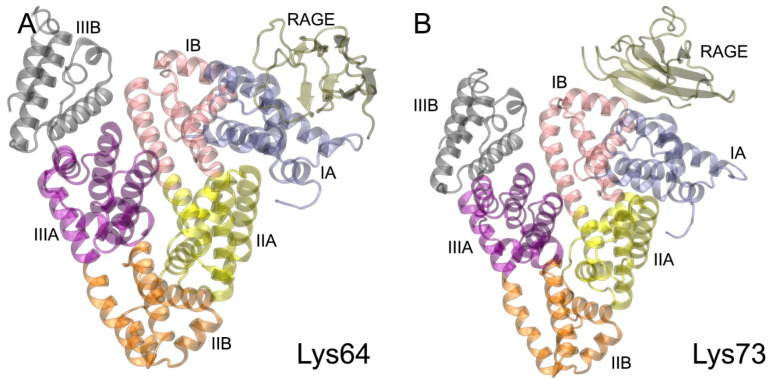
The MD structures of the complexes of RAGE with gHSA glycated at Lys64 (**A**), Lys73 (**B**), Lys137 (**C**), Lys233 (**D**), Lys262 (**E**), Lys317 (**F**), Lys378 (**G**), Lys525 (**H**), Lys573 (**I**) and Lys574 (**J**). Domains IA, IB, IIA, IIB, IIIA, and IIIB of gHSA are shown with ice blue, pink, yellow, orange, purple and black ribbon, respectively. The V-domain of RAGE is shown with brown ribbon.

**Figure 7 ijms-25-03204-f007:**
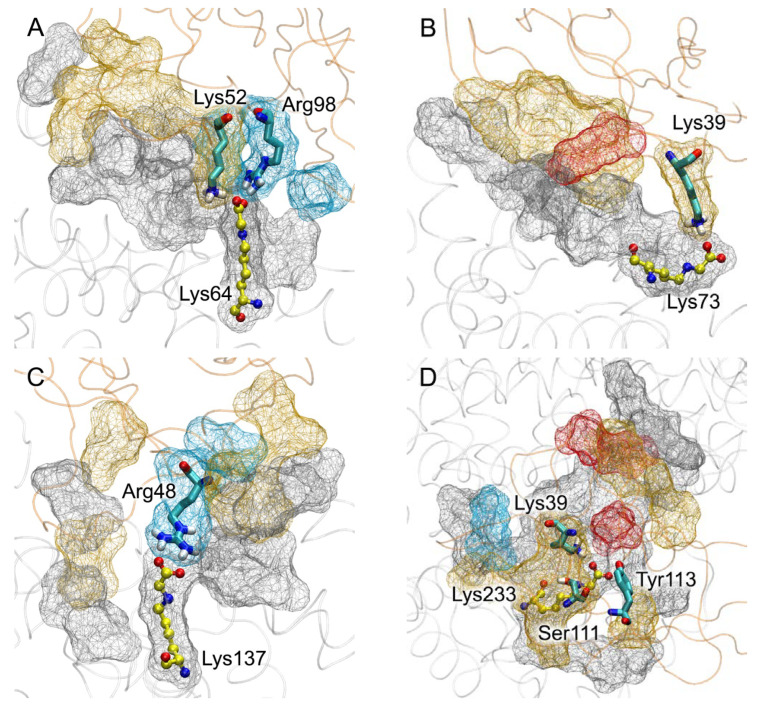
Interaction of RAGE with gHSA glycated at Lys64 (**A**), Lys73 (**B**), Lys137 (**C**), Lys233 (**D**), Lys262 (**E**), Lys317 (**F**), Lys378 (**G**), Lys525 (**H**), Lys573 (**I**) and Lys574 (**J**) according to MD simulation. The backbones of the HSA and the V-domain of RAGE are shown as the gray and orange threads, respectively. The regions of gHSA contacting with RAGE are shown as a gray surface. The regions of RAGE contacting with gHSA are shown with red (IS1), blue (IS2), green (IS3) or brown (none of the IS) surfaces. Carbon, hydrogen, oxygen and nitrogen atoms of amino acids are shown in cyan, white, red and blue, respectively. The glycated lysines of gHSA are shown in the “ball and sticks” representation; their carbon atoms are highlighted in yellow. The residues of RAGE that are involved in the interaction with CML of gHSA are shown in the “sticks” representation. Nonessential hydrogens are not shown for clarity. For clarity, the projections of the images presented in this figure do not always correspond to the projections in Figure 4, Figure 5 and Figure 6.

**Figure 8 ijms-25-03204-f008:**
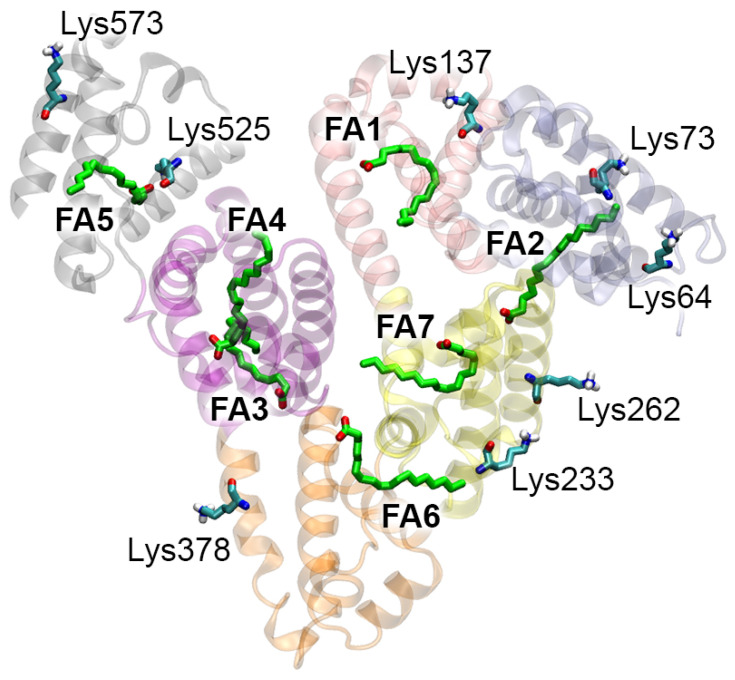
Topology of the HSA molecule: oleic acid (OLA) molecules bound in the fatty acid (FA) binding sites FA1–7 according to PDB entry 1GNI [31]) and the lysine residues that play a key role in the binding of gHSA to RAGE. Domains IA, IB, IIA, IIB, IIIA and IIIB of albumin are highlighted in ice blue, pink, yellow, orange, purple and black, respectively. Carbon, hydrogen, oxygen and nitrogen atoms are shown in cyan, white, red and blue, respectively; carbon atoms of OLA are highlighted in green.

**Figure 9 ijms-25-03204-f009:**
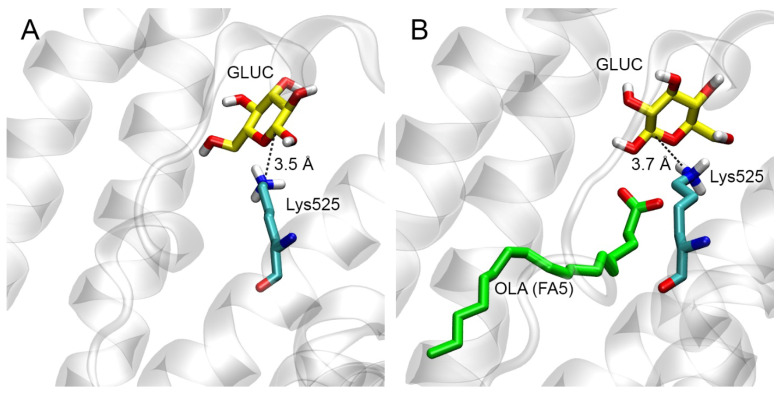
The result of molecular docking of β-D-glucopyranose (GLUC) near Lys525 in the absence (**A**) and presence (**B**) of the OLA molecule in site FA5. Carbon, hydrogen, oxygen and nitrogen atoms are shown in cyan, white, red and blue, respectively; carbon atoms of OLA and GLUC are highlighted in green and yellow, respectively; nonessential hydrogens are omitted for clarity.

**Figure 10 ijms-25-03204-f010:**
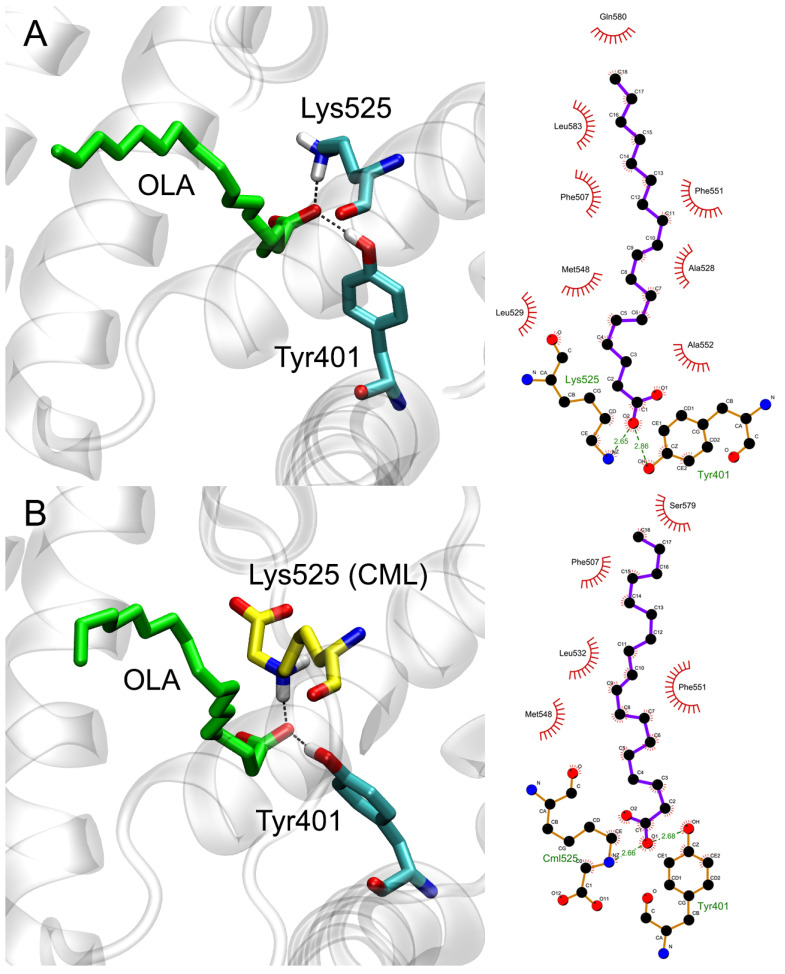
Conformation of the OLA molecule in site FA5 of native HSA (**A**) and gHSA glycated at Lys525 (**B**). The left and right panels show the 3D and 2D representation, respectively. Carbon, hydrogen, oxygen and nitrogen atoms of amino acids are shown in cyan, white, red and blue, respectively; carbon atoms of OLA and CML are highlighted in green and yellow, respectively. Nonessential hydrogens are omitted for clarity.

**Figure 11 ijms-25-03204-f011:**
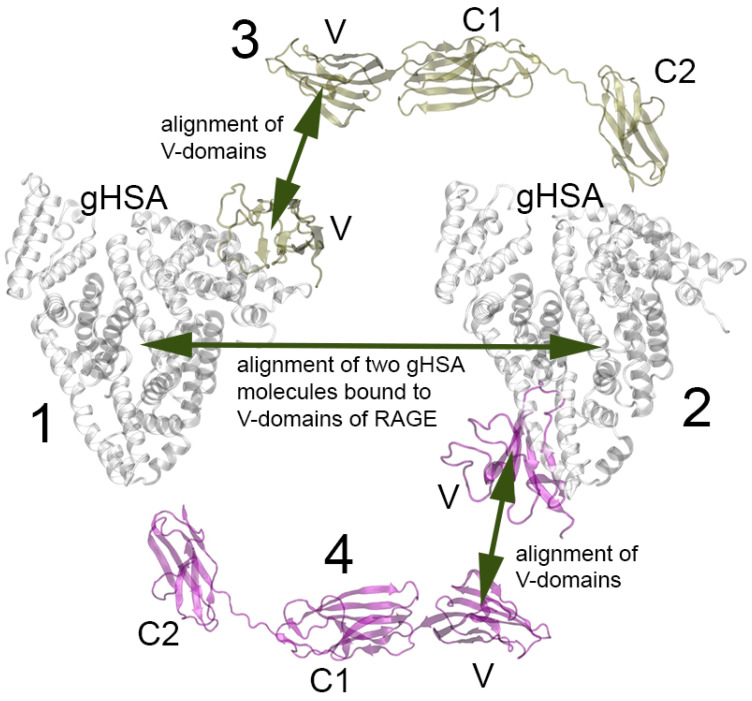
Scheme of the alignment of HSA and RAGE molecules to obtain the structure of the dimers. Symbols 1 and 2 indicate complexes of gHSA with the V-domain of RAGE, obtained in silico in the presented work. Symbols 3 and 4 indicate the structures of RAGE VC1C2-domains obtained experimentally (PDB entry 4LP5, chain A [12]). The symbols V, C1 and C2 indicate the V-, C1- and C2-domains of RAGE, respectively.

**Figure 12 ijms-25-03204-f012:**
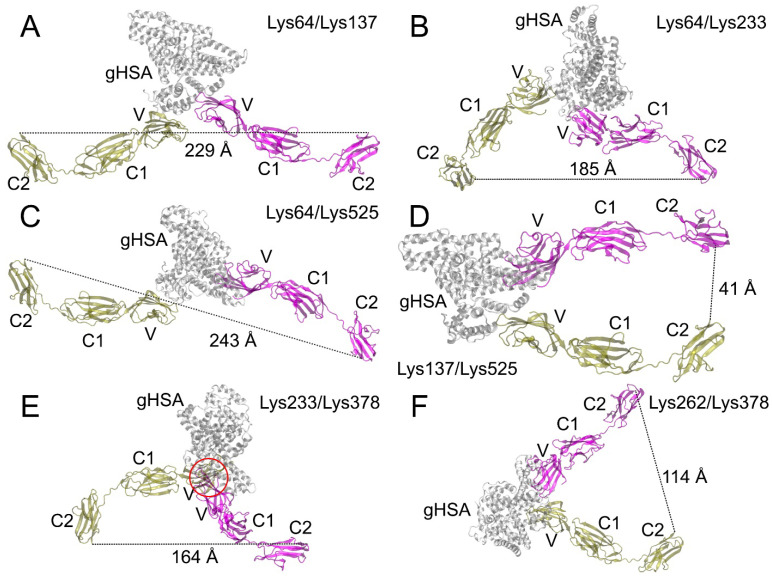
Possible structures of the RAGE dimers in complex with gHSA according to molecular modeling. Models of gHSA are presented in which the following pairs of lysines are glycated: Lys64/Lys137 (**A**), Lys64/Lys233 (**B**), Lys64/Lys525 (**C**), Lys137/Lys525 (**D**), Lys233/Lys378 (**E**), Lys262/Lys378 (**F**). The symbols V, C1 and C2 indicate the V-, C1- and C2-domains of RAGE, respectively. Red circle indicates the overlapping of V-domains.

**Figure 13 ijms-25-03204-f013:**
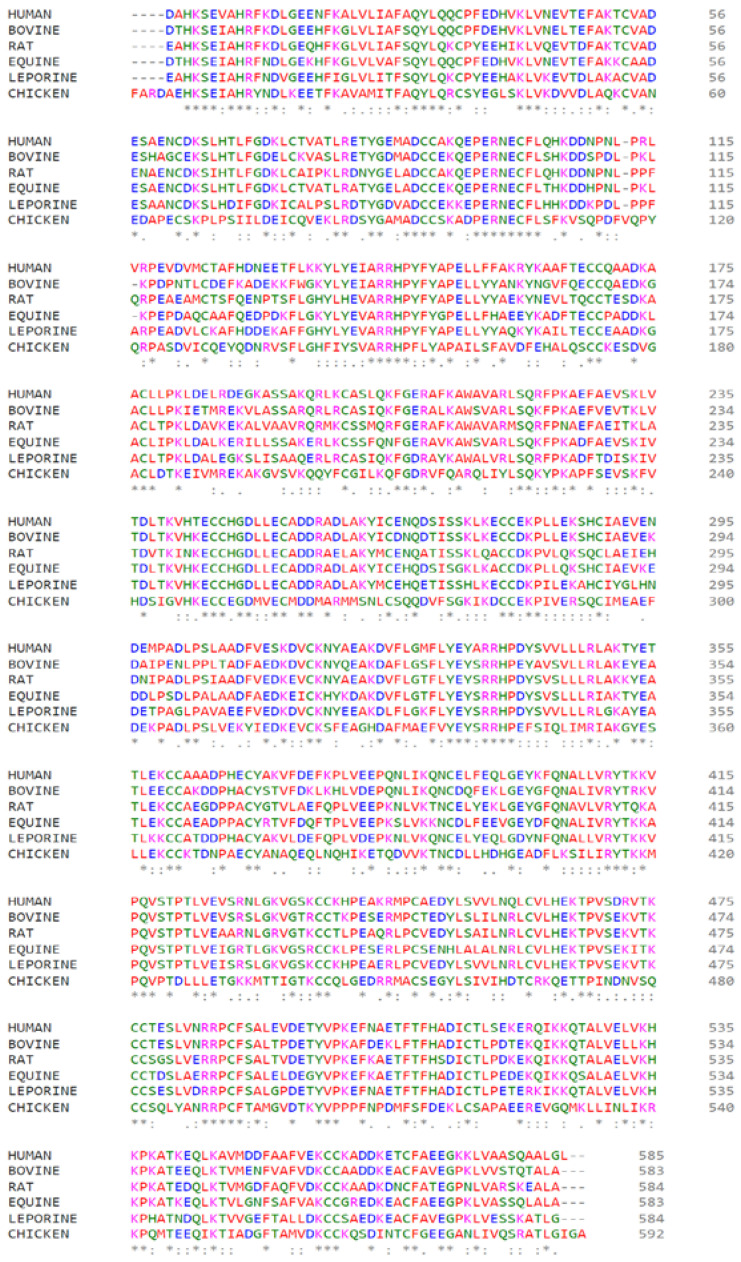
Alignment of the primary sequences of albumins from different species. “*”, the residues are identical in all the sequences in the alignment; “:”, conserved substitutions are observed; “.”, semi-conserved substitutions are observed; “spacebar”, non-conserved substitutions are observed.

**Figure 14 ijms-25-03204-f014:**
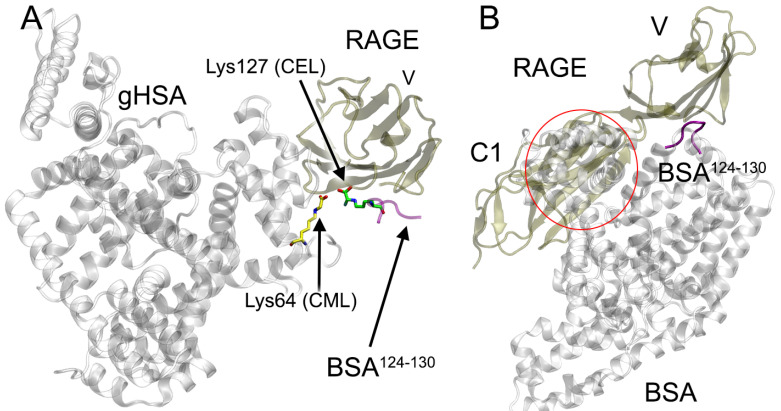
Interaction of RAGE with glycated lysines within AGE proteins and AGE peptides according to in vitro and molecular modeling data. (**A**), the superimposed structures of the complex of RAGE with Lys64-glycated gHSA obtained by molecular modeling in the present work and the complex of RAGE with Lys127-glycated fragment of bovine serum albumin (residues 124–130, BSA^124−130^) obtained by NMR (PDB entry 2L7U [55]). The carbon atoms of modified Lys64 of gHSA are highlighted in yellow; the carbon atoms of modified lysine (carboxyethyl-lysine, CEL) of BSA^124−130^ are highlighted in green. The backbone of the gHSA, RAGE and peptide BSA^124−130^ is shown as gray, brown and purple ribbon, respectively. Nonessential hydrogens are not shown for clarity. (**B**), the superimposed structures of the full-length BSA molecule (PDB entry 6QS9 [35]), the VC1-domains of RAGE (PDB entry 4OI8 [15]) and the complex of RAGE with BSA^124−130^ (PDB entry 2L7U [55]). The backbone of BSA, RAGE and peptide BSA^124−130^ is shown as gray, brown and purple ribbon, respectively. The symbols V and C1 indicate the V- and C1-domain of RAGE, respectively. The overlap of BSA and the C1-domain is outlined in red.

**Table 1 ijms-25-03204-t001:** Experimentally obtained three-dimensional structures of the complexes of RAGE with its ligands.

PDB Entry	Structure Determination Method	Description of the Structure	Reference
2M1K	NMR	Dimer of the V-domain of RAGE in complex with dimer of protein S100A6	[13]
2MJW	NMR	Dimer of the V-domain of RAGE in complex with dimer of protein S100P	[14]
4OI7 and 4OI8	X-Ray	Dimer of the VC1-domains of RAGE in complex with DNA	[15]

**Table 2 ijms-25-03204-t002:** Scoring function (SF) of conformations of the RAGE complexes with gHSA glycated at different lysine residues, according to macromolecular docking.

Glycated Lys of gHSA	ZDOCK Score
Lys64	741.0
Lys73	888.3
Lys137	1011.2
Lys233	1059.3
Lys262	963.0
Lys317	1034.8
Lys378	878.7
Lys525	828.5
Lys573	895.5
Lys574	828.6

**Table 3 ijms-25-03204-t003:** MD simulation of gHSA complexes with RAGE: the number of atoms forming short-range (3.5 Å) contacts; the main residues involved in the interaction between gHSA and RAGE; the specific gHSA-RAGE contacts (salt bridges, hydrogen bonds and π-π interactions); and their lifetime as a percentage of the simulation length.

Glycated Lys of gHSA	gHSA	RAGE	Specific Contacts gHSA-RAGE
Lys64	63 Glu48, Glu45, Lys64, Asp72, Thr76, Val77, Glu86, Cys91, Val92, Lys93, Gln94, Glu95	73 Arg48, Lys52, Glu59, Ala60, Trp61, Lys62, Arg98, Lys107	Glu45-Arg48 (SB, 100%); Glu48-Lys107 (SB, 30%); Lys64-Lys52 (SB, 80%); Lys64-Arg98 (SB, 95%); Asp72-Lys52 (SB, 80%); Glu86-Lys62 (SB, 50%); Glu95-Lys52 (SB, 70%)
Lys73	66 Glu37, Asp38, Val40, Lys41, Glu45, Lys73, Glu82, Val122, Thr125, Ala126, Asp129, Thr133	68 Gln24, Asn25, Lys37, Lys39, Arg114, Arg116	Glu37-Gln24 (HB, 40%); Asp38-Lys37 (SB, 40%); Glu45-Lys39 (SB, 65%); Lys73-Lys39 (SB, 30%); Glu82-Lys37 (SB, 40%); Asp129-Arg114 (SB, 85%); Asp129-Arg116 (SB, 60%)
Lys137	68 Glu37, Pro35, Asp38, Glu82, Thr83, Tyr84, Arg114, Val122, Thr125, Lys137	58 Lys44, Gln47, Arg48, Gln67, Met102, Arg104, Asn105, Glu108,	Glu37-Arg48 (SB, 5%); Asp38-Arg104 (SB, 50%); Asp38-Gln47 (HB, 10%); Glu82-Lys44 (SB, 5%); Glu83-Asn105 (HB, 5%); Arg114-Glu108 (SB, 20%); Lys137-Arg48 (SB, 95%)
Lys233	104 His3, Lys4, Glu6, His9, Glu208, Lys212, Glu230, Lys233, Thr236, Asp237, Lys240, Glu252, Asp255, Asp256, Asp259, Lys262, Glu266, Asn267	103 Gln24, Val35, Lys37, Lys39, Lys43, Leu79, Pro80, Asn81, Lys107, Glu108, Thr109, Lys110, Ser111, Tyr113	His3-Asn81 (HB, 95%); Glu6-Lys83 (SB, 95%); Glu208-Lys107 (SB, 95%); Lys212-Glu108 (SB, 75%); Glu230-Lys110 (SB, 20%); Lys233-Lys39 (SB, 80%); Lys233-Ser111 (HB, 65%); Lys233-Tyr113 (HB, 40%); Thr236-Glu108 (HB, 60%); Asp237-Lys39 (SB, 80%); Glu252-Lys43 (SB, 98%); Asp255-Lys37 (SB, 60%); Asp256-Lys37 (SB, 60%); Asp259-Lys37 (SB, 60%)
Lys262	65 Asp13, Lys233, Thr236, Asp237, Glu252, Asp255, Asp256, Asp259, Lys262, Tyr263, Glu266	54 Lys39, Lys43, Lys44, Asn105, Gly106, Lys107, Glu108	Asp13-Lys43 (SB, 90%); Lys233-Glu108 (SB, 90%); Thr236-Asn105 (HB, 90%); Glu252-Lys44 (SB, 70%); Asp255-Lys43 (SB, 90%); Asp256-Lys44 (SB, 75%); Asp259-Lys107 (SB, 10%); Lys262-Lys39 (SB, 10%); Tyr263-Glu108 (HB, 90%); Glu266-Lys39 (SB, 70%)
Lys317	21 Ala229, Glu230, Lys233, Asp259	27 Gln24, Lys39, Lys107, Ser111	Glu230-Lys39 (SB, 2%); Lys233-Ser111 (HB, 10%); Asp259-Lys107 (SB, 70%)
Lys378	116 Glu297, Ala300, Pro303, Ala306, Ala307, Glu311, Asp340, Phe374, Asp375, Lys378, Val381, Gln385, Pro441, Glu442, Met446	109 Arg29, Glu32, Pro33, Leu34, Val35, Lys37, Lys39, Pro46, Arg48, Arg77, Val78, Pro80, Asn81, Phe85	Glu297-Lys39 (SB, 50%); Glu311-Arg29 (SB, 80%); Phe374-Phe85 (PP, 20%); Asp375-Arg77 (SB, 95%); Lys378-Arg77 (SB, 20%); Gln385-Pro80 (HB, 60%); Glu442-Arg48 (SB, 30%)
Lys525	101 Glu119, Asp173, Ala175, Ala176, Leu179, Pro180, Asp183, Glu184, Arg186, Glu518, Lys519, Glu520, Lys525, Gln522, Lys560, Asp562	109 Lys39, Gly40, Pro42, Lys44, Arg48, Lys52, Arg57, Glu59, Met102, Arg104, Asn105, Gly106, Lys107, Glu108, Thr109	Glu119-Lys107 (SB, 80%); Asp173-Lys39 (SB, 90%); Asp183-Asn105 (HB, 60%); Glu184-Lys44 (SB, 65%); Glu518-Lys52 (SB, 80%); Lys519-Glu108 (SB, 90%); Glu520-Arg104 (SB, 80%); Lys525-Arg48 (SB, 5%); Asp562-Arg57 (SB, 90%)
Lys573	47 Pro113, Arg114, Leu115, Val116, Ala511, Asp512, Glu565, Thr566, Ala569, Lys573	59 Arg29, Pro33, Pro66, Gln67, Arg77, Leu79, Phe85	Val116-Pro66 (HB, 5%); Asp512-Arg77 (SB, 50%); Lys573-Arg29 (SB, 30%); Glu565-Arg77 (SB, 70%)
Lys574	142 Phe36, Glu37, Tyr84, Leu112, Arg114, Leu115, Val116, Asp121, Val122, Thr125, Asp129, Lys137, Tyr140, Glu505, Thr508, His510, Ala511, Glu565, Phe568, Ala569, Lys573	139 Gly31, Glu32, Pro33, Val35, Lys37, Lys43, Pro45, Pro46, Arg48, Gln67, Ser74, Arg77, Leu79, Pro80, Phe85, Arg104	Asp129-Arg104 (SB, 80%); Asp121-Lys43 (SB, 20%); Glu37-Arg48 (SB, 60%); Glu565-Lys37 (SB, 90%); Glu505-Arg77 (SB, 70%); Phe36-Gln67 (HB, 30%); Tyr140-Gln67 (HB, 60%); Lys573-Gly31 (HB, 80%); Arg114-Ser74 (HB, 10%)

The modified lysines (CML) of gHSA and the RAGE residues that interact with CML are indicated in underlined font. The residues belonging to experimentally identified sites of interaction of RAGE with AGE peptides and gBSA [10] are indicated in colored font: IS1 (red), IS2 (blue) and IS3 (green). SB, salt bridge; HB, hydrogen bond; PP, π–π interaction.

**Table 4 ijms-25-03204-t004:** Distances (Å) between the C2-domain monomers in the constructed complexes of the RAGE dimers with gHSA.

	Lys64	Lys137	Lys233	Lys262	Lys378	Lys525	Lys573
Lys64	–	–	–	–	–	–	–
Lys137	229	–	–	–	–	–	–
Lys233	185	191	–	–	–	–	–
Lys262	188	213	24 *	–	–	–	–
Lys378	176	243	164 *	114	–	–	–
Lys525	243	41	191	213	228	–	–
Lys573	116	162 *	223	238	227	177	–

*, the monomers of the V-domain overlap in these complexes, which means that such a complex cannot be formed.

**Table 5 ijms-25-03204-t005:** Variability of the gHSA amino acids involved in the interaction with RAGE.

Glycated Lys of gHSA	Major gHSA Amino Acids Involved in the Interaction with RAGE	Variability in Mammals (HSA, BSA, RSA, ESA, LSA)	Variability in Mammals and Chicken (HSA, BSA, RSA, ESA, LSA, CSA)
Lys64	Glu45	*	:
	Glu48	:	:
	Lys64	*	*
	Asp72	*	*
	Glu86	:	-
	Glu95	*	*
Lys233	His3	*	*
	Glu6	*	*
	Glu208	:	:
	Lys212	*	:
	Glu230	:	:
	Lys233	*	*
	Thr236	*	-
	Asp237	*	*
	Glu252	*	*
	Asp255	*	*
	Asp256	*	*
	Asp259	*	-
Lys262	Asp13	*	*
	Lys233	*	*
	Thr236	*	-
	Glu252	*	*
	Asp255	*	*
	Asp256	*	*
	Asp259	*	-
	Lys262	*	.
	Tyr263	*	-
	Glu266	:	.
Lys378	Glu297	-	-
	Glu311	*	*
	Phe374	-	-
	Asp375	-	-
	Lys378	-	-
	Gln385	:	:
	Glu442	*	*
Lys525	Glu119	:	-
	Asp173	*	*
	Asp183	:	-
	Glu184	-	-
	Glu518	:	-
	Lys519	-	-
	Glu520	*	*
	Lys525	*	:
	Asp562	-	-

“*”, the residues are identical in all the sequences in the alignment; “:”, conserved substitutions are observed; “.”, semi-conserved substitutions are observed; “-”, non-conserved substitutions are observed; HSA, BSA, RSA, ESA, LSA and CSA, human, bovine, rat, equine, leporine and chicken serum albumin, respectively; other abbreviations are the same as in Table 3.

**Table 6 ijms-25-03204-t006:** Lysine composition of albumins of different species.

Albumin	Total Number of Lysines	Total Number of Lysines on the Protein Surface Available for Interaction with RAGE
HSA	59	43
BSA	59	47
RSA	53	39
ESA	59	50
LSA	57	48
CSA	47	36

HSA, BSA, RSA, ESA, LSA and CSA, human, bovine, rat, equine, leporine and chicken serum albumin, respectively.

## Data Availability

The data presented in this study are available on request.

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
