# Peer review of "Molecular Basis for the Involvement of Mammalian Serum Albumin in the AGE/RAGE Axis: A Comprehensive Computational Study"

_ijms, 2024, doi:10.3390/ijms25063204_

Round 1

Reviewer 1 Report

Comments and Suggestions for Authors

Comments for Authors

The current manuscript entitled “Molecular basis for the involvement of mammalian serum albumin in the AGE/RAGE axis: a comprehensive computational study” presented the interaction of glycated human albumin (gHSA) with RAGE using molecular modeling methods. The current article will contribute new insights into the subject through a synthesis of knowledge from the recent literature. While the inclusion of some general contextual content may be appropriate, the structure and style of the writing should strike a clear balance between breadth and depth. To me, the main innovation point of this research is to investigate the interaction of gHSA with RAGE using molecular modeling methods and perform multiple macromolecular docking between gHSA and RAGE. The concept of the study is good and the manuscript is publishable before some minor changes are suggested;

The abstract is one of the most important parts of a research article. According to journal guidelines, the abstract should be a total of about 200 words maximum. However, the current abstract contains 343 words. Use short punchy statements in the abstract section.

Line 36, It is more transparent if you use alphabetical order of some of the used keywords. For example: Carboxymethyl-lysine, diabetes mellitus; fatty acids; glycated albumin; molecular modeling; and receptor for advanced glycation end products.

Line 39, (AGE) should be changed with (AGEs).

The introduction section could be reorganized. Some parts must be reduced so more focus can be given to diabetes mellitus, the accumulation of advanced glycation end products (AGEs) leads to inflammation and oxidative stress through the activation of specific receptors for AGE (RAGE). For instance, consult the following study and improve the introduction section

https://link.springer.com/article/10.1134/S0022093023060285

In lines 77-82, the following sentences Until three-dimensional structure ……. in mammals.” need to be rewritten to make better sense of what needs to be studied.

Line 224, Figures (Figure S3C–J), Figure S2H–J), and (Figure S2A–G) are cited in the text. However, these figures are not present in the main file or supplementary material.

In lines 305-309, the statement “Therefore, for protein-protein complexes, …...” should be rephrased for clarity.

Line 376, “one pi-pi interaction” changes with “π-π interaction”.

Line 505, Rephrase the following sentence “We performed molecular docking of …..” for clarity. free HSA and HAS???  

Line 850: In general, the limitations of the study are stated at the beginning of the discussion section. Having read the limitations, the reader is already aware of these limitations and can proceed to read the rest of your analysis.

It is recommended to proofread the current manuscript to correct typos, grammatical errors, and syntactical errors.

Comments on the Quality of English Language

Moderate editing of the English language is required to improve the quality of the current manuscript. It is recommended to proofread the current manuscript to correct typos, grammatical errors, and syntactical errors.

Reviewer 2 Report

Comments and Suggestions for Authors

While the AGE/RAGE interactions seem to be quite important for human
health, I do not believe much was learned in the study described in
this manuscript. The study runs up against limitations in docking of
structures which may or may not be relevant to disease.

The fact that no structures were deposited or otherwise made
available, coupled with a missing ``Appendix'' warrant automatic
rejection of the manuscript. If the result of the molecular dynamics
and docking calculations are structures, these must be deposited in a
public repository.

Moreoever, little justification was given to the approach taken here:
why were only singly-glycated albumin studied? The dimers contained
only monomers glycated at different sites. Studying this system is
indeed a difficult heterogeneous problem, as previous studies have
identified glycation on most of the 59 lysine and 24 arginine
residues of the serum albumin protein. There is also the possibility
that glycation is accompanied by other post-translational
modifications.

In short, this reviewer does not believe that the glycation+docking
procedure used here is capable of producing reliable results.
In approaching this problem, I would first perform some subset of
calculations which could be verifed experimentally before running a
zoo of calculations which likely lead farther and farther from
reality. For instance, there seems a high likelihood that dimers of
glycated albumin and RAGE interact. Then, why study docking of the
monomers? Could any connection to experiment be drawn from the
results?

Finally, the authors should not refer to their computational results
as ``data''.

Author Response

We thank Reviewer-2 for valuable comments on the manuscript. We have substantially modified the manuscript to address the points raised. The changes are detailed below and highlighted by green in the revised and improved manuscript.

---------------------------------------------------------------------------------------------------------------------

While the AGE/RAGE interactions seem to be quite important for human health, I do not believe much was learned in the study described in this manuscript. The study runs up against limitations in docking of structures which may or may not be relevant to disease.

Indeed, the method of molecular docking has its limitations, which we have described in the “Limitations” section. However, in the presented paper we performed the molecular docking procedure based on known experimental data. For MD simulation, we choose those complexes in which albumin interacted with “IS” sites on the surface of RAGE. “IS” sites (IS1, IS2 and IS3) were determined experimentally [Xie et al., 2008 https://doi.org/10.1074/jbc.m801622200] as the sites of interaction of RAGE with glycated lysine within AGE-peptides and glycated bovine serum albumin (BSA).

We have also expanded the Discussion section to include information about the significance of the results obtained. It is known that HSA is a transport protein for many endogenous and exogenous ligands; the protein molecule contains three major and several minor ligand-binding centers. Moreover, albumin is liable to allosteric modulation (binding of a ligand in one site change the binding activity of other sites). Firstly, the information obtained on the structure of gHSA complexes with RAGE seems useful to us from the point of view of how interaction with RAGE will change the affinity of the protein for various pharmaceuticals. Secondly, albumin's susceptibility to allosteric modulation may be a key for management of its interaction with RAGE.

---------------------------------------------------------------------------------------------------------------------

The fact that no structures were deposited or otherwise made available, coupled with a missing ``Appendix'' warrant automatic rejection of the manuscript. If the result of the molecular dynamics and docking calculations are structures, these must be deposited in a public repository.

The docking and MD structures are uploaded in PDB format as Appendix A and B, respectively.

The supplementary material was downloaded during the initial submission of the manuscript and is downloaded now (Appendix C); it contains all the pictures that are mentioned in the article. Perhaps there was some error during the download, we have now duplicated the Appendix in a  letter to the editor.

---------------------------------------------------------------------------------------------------------------------

Moreoever, little justification was given to the approach taken here: why were only singly-glycated albumin studied? The dimers contained only monomers glycated at different sites. Studying this system is indeed a difficult heterogeneous problem, as previous studies have identified glycation on most of the 59 lysine and 24 arginine residues of the serum albumin protein.

The goal of our wider study (which is not limited to the presented paper) is to determine the molecular basis for the involvement of albumin in the AGE/RAGE axis. Therefore, we decided to work sequentially and to study mono-glycated albumin first, and in the subsequent stages to complicate the task by considering multiple glycations. The advantage of molecular modeling is that it can be used to study the contribution of each glycation separately (which cannot be achieved by in vitro and in vivo experiments). Determination of single glycation contribution is important for comparing albumins of different types, for developing DM therapy (since glycation and/or binding to RAGE can affect the affinity of the protein for various pharmaceuticals), and for studying the phenotypic features of patients with mutated albumin (for example, the Canterbury 313Lys→Asn, Vanves 574Lys→Asn or Verona mutation 570Glu→Lys). We have added this rationale to the article (beginning of section 2.2).

---------------------------------------------------------------------------------------------------------------------

There is also the possibility that glycation is accompanied by other post-translational modifications.

Indeed, glycation of albumin is often accompanied by oxidation of the thiol group of Cys34. Firstly, oxidation can affect the conformational and binding characteristics of albumin, and secondly, one of the forms of oxidized HSA is a dimer through the Cys34-Cys34 disulfide bridge. However, in the Discussion and Limitations sections, we have discussed that the effect of oxidation was not considered in the presented work, and will be considered in subsequent articles.

---------------------------------------------------------------------------------------------------------------------

In short, this reviewer does not believe that the glycation+docking procedure used here is capable of producing reliable results. In approaching this problem, I would first perform some subset of calculations which could be verifed experimentally before running a zoo of calculations which likely lead farther and farther from reality.

The result of our calculations is not completely divorced from reality. As noted in our response to the first comment, we did not select the results of molecular docking completely blindly. We chose those complexes in which albumin interacted with “IS” sites on the surface of the V-domain. “IS” sites (IS1, IS2 and IS3) were determined experimentally [Xie et al., 2008 https://doi.org/10.1074/jbc.m801622200] as the sites of interaction of RAGE with glycated lysine within AGE-peptides and glycated bovine serum albumin (BSA).

---------------------------------------------------------------------------------------------------------------------

For instance, there seems a high likelihood that dimers of glycated albumin and RAGE interact. Then, why study docking of the monomers? Could any connection to experiment be drawn from the results?

Previously, it was established that at physiologically relevant concentrations HSA does form weak, reversible non-covalent dimers [Chubarov et al., 2020 https://doi.org/10.3390/molecules26010108]. Presumably, in the bloodstream and in extravascular fluids, there is some equilibrium between the concentration of albumin dimers and monomers. Since extravascular fluids contain much lower levels of HSA, the equilibrium is shifted toward more monomers and fewer dimers [Chubarov et al., 2020 https://doi.org/10.3390/molecules26010108]. Since albumin interacts with RAGE both in blood vessels and in extravascular fluids, it seems reasonable to devote the presented study to the binding of albumin monomers to RAGE, and then, in future works, to model the binding of the dimers on the basis of the described information.

We have added this rationale to the article (beginning of section 2.2).

---------------------------------------------------------------------------------------------------------------------

Finally, the authors should not refer to their computational results as ``data''.

Corrected

Round 2

Reviewer 2 Report

Comments and Suggestions for Authors

The updated manuscript is improved in that it provides structures for the monomers, and some additional rationale. It should be noted that structures of the dimers are still missing.

However, to this reviewer, this work does not constitute a publishable unit: my understanding is that ZDOCK gets the qualitatively correct answer about half of the time. So the resulting structures are useful to generate and test hypotheses, or to help understand or rationalize results. If these structures somehow help you design experiments which lead to their validation or generate inhibitors or other therapy, then the results can be published, but they do not stand on their own as a representation of reality.

Author Response

We thank Reviewer-2 for valuable comments on the manuscript. We have substantially modified the manuscript to address the points raised. The changes are detailed below and highlighted by green in the revised and improved manuscript.

-------------------------------------------------------------------------------------------------------------------

The updated manuscript is improved in that it provides structures for the monomers, and some additional rationale. It should be noted that structures of the dimers are still missing.

As it was mentioned in Section 2.6, we consider the Lys64-Lys233, Lys64-Lys262 and Lys262-Lys378 dimers to be the leading ones. In the revised version of the manuscript, the structures of these dimers optimized by energy minimization using GROMACS 2019.4 are available in PDB format in the Supplementary materials (Appendix D). We also changed the order of the Appendices according to their first mention in the text.

-------------------------------------------------------------------------------------------------------------------

However, to this reviewer, this work does not constitute a publishable unit: my understanding is that ZDOCK gets the qualitatively correct answer about half of the time. So the resulting structures are useful to generate and test hypotheses, or to help understand or rationalize results. If these structures somehow help you design experiments which lead to their validation or generate inhibitors or other therapy, then the results can be published, but they do not stand on their own as a representation of reality.

We greatly appreciate this opinion and the principled position of the respected reviewer. For our part, we are confident in the high quality of the work carried out, which is in no way inferior to numerous publications of results obtained in silico in this and some other journals of the Q1 category. By the way, many of this kind of publications limit their studies to molecular docking without molecular dynamics, all other things being equal. This in no way means that we do not want and do not plan to conduct experiments in vitro, ex vivo or in vivo; just preparing and conducting such experiments takes time and requires additional financial investments, but this work is a priority for us in this year.

Thank you for understanding.